# Reversible phosphorylation of cyclin T1 promotes assembly and stability of P-TEFb

**Fang Huang[1,2], Trang TT Nguyen[1,2,3], Ignacia Echeverria[4,5], Ramachandran Rakesh[4,5], Daniele C Cary[1,2], Hana Paculova[1], Andrej Sali[4,6], Arthur Weiss[1,2,3], Boris Matija Peterlin[1,2]\*, Koh Fujinaga[1,2]\***

[1]Departments of Medicine, Microbiology and Immunology, University of California, San Francisco, San Francisco, United States; [2]Department of Medicine, San Francisco, United States; [3]The Howard Hughes Medical Institute, San Francisco, United States; [4]Department of Bioengineering and Therapeutic Sciences, University of California, San Francisco, San Francisco, United States; [5]Departmentof Cellular Molecular Pharmacology, California Institute for Quantitative Biosciences (QBI), and Department of Bioengineering and Therapeutic Sciences, San Francisco, United States; [6]Department of Bioengineering and Therapeutic Sciences, Department of Pharmaceutical Chemistry, and California Institute for Quantitative Biosciences (QBI), San Francisco, United States

**\*For correspondence:**
Matija.Peterlin@ucsf.edu (BMatijaP);
koh.fujinaga@ucsf.edu (KF)

**Competing interest:** The authors declare that no competing interests exist.

**Abstract** The positive transcription elongation factor b (P-TEFb) is a critical coactivator for transcription of most cellular and viral genes, including those of HIV. While P-TEFb is regulated by 7SK snRNA in proliferating cells, P-TEFb is absent due to diminished levels of CycT1 in quiescent and terminally differentiated cells, which has remained unexplored. In these cells, we found that CycT1 not bound to CDK9 is rapidly degraded. Moreover, productive CycT1:CDK9 interactions are increased by PKC-mediated phosphorylation of CycT1 in human cells. Conversely, dephosphorylation of CycT1 by PP1 reverses this process. Thus, PKC inhibitors or removal of PKC by chronic activation results in P-TEFb disassembly and CycT1 degradation. This finding not only recapitulates P-TEFb depletion in resting CD4+ T cells but also in anergic T cells. Importantly, our studies reveal mechanisms of P-TEFb inactivation underlying T cell quiescence, anergy, and exhaustion as well as proviral latency and terminally differentiated cells.

## Editor's evaluation

This study addresses an important question regarding the regulation of positive transcription elongation factor b (P-TEFb), which is an important regulator of gene expression targeted by the HIV Tat protein. The authors propose a novel mechanism with the potential to explain the differences in P-TEFb regulation between proliferating and quiescent cells, which might in turn have important implications for antiviral therapy.

## Introduction

In eukaryotic cells, coding gene expression starts with transcription of DNA to RNA by RNA polymerase II (RNAPII) in the nucleus. This process consists of initiation, promoter clearance, capping, elongation, and termination upon which the transcribed single-strand RNA is cleaved and polyadenylated before transport to the cytoplasm (*Lis, 2019*; *Peterlin and Price, 2006*; *Proudfoot,*

*2016*; *Zhou et al., 2012*). Clearance of RNAPII from promoters requires the phosphorylation of its C-terminal domain (CTD) at position 5 (Ser5) in tandemly repeated heptapeptide (52 repeats of Tyr1-Ser2-Pro3-Thr4-Ser5-Pro6-Ser7) by cyclin-dependent kinase 7 (CDK7) from the transcription factor-II H (TFIIH) (*Chapman et al., 2008*). Of interest, promoters of most inactive and inducible genes are already engaged by stalled RNAPII, which departs from the transcription start site (TSS) but pauses after transcribing 20–100-nucleotides-long transcripts (*Rahl et al., 2010*).

The pause of RNAPII is caused by two factors, the negative elongation factor (NELF) (*Yamaguchi et al., 1999*) and DRB sensitivity-inducing factor (DSIF) (*Wada et al., 1998a*). The release of RNAPII for productive elongation requires the positive transcription elongation factor b (P-TEFb), which is composed of cyclin-dependent kinase 9 (CDK9) and C-type cyclins T1 or T2 (CycT1 or CycT2) (*Zhou et al., 2012*). Compared to the largely restricted expression of CycT2, CycT1 is expressed ubiquitously (*Peng et al., 1998*). N-terminus of CycT1 contains two highly conserved cyclin boxes for CDK9 binding, followed by the Tat-TAR recognition motif (TRM), a coil-coiled motif, the histidine (His)-rich motif that binds to the CTD, and a C-terminal PEST motif (*Taube et al., 2002*; *Wei et al., 1998*). CDK9 is a Ser/Thr proline-directed kinase (i.e., PITALRE) (*Graña et al., 1994*) that phosphorylates Spt5 in DSIF and NELF-E, which relieve the pausing of RNAPII (*Fujinaga et al., 2004*; *Ivanov et al., 2000*). After phosphorylation, NELF is released and DSIF is converted to an elongation factor (*Wada et al., 1998b*). Serine at position 2 (Ser2) in the CTD is also phosphorylated by CDK9 before RNAPII's transition to productive elongation (*Peterlin and Price, 2006*). Thus, P-TEFb is a critical factor for transcriptional elongation and co-transcriptional processing by RNAPII.

In the organism, the kinase activity of P-TEFb is kept under a tight control to maintain the appropriate state of growth and proliferation of cells. To ensure this balance, a large complex, known as the 7SK small nuclear ribonucleoprotein (7SK snRNP), sequesters a large amount of P-TEFb (from 50 to 90% in different cells) in an inactive state (*Peterlin et al., 2012*). 7SK snRNP consists of the abundant 7SK small nuclear RNA (7SK snRNA), hexamethylene bisacetamide (HMBA)-inducible mRNAs 1 and 2 (HEXIM1/2) proteins, La-related protein 7 (LARP7), and methyl phosphate capping enzyme (MePCE) (*Michels and Bensaude, 2008*; *Zhou et al., 2012*). Proper levels and activities of P-TEFb ensure appropriate responses to external stimuli. They also maintain states of differentiation, growth, and proliferation of cells (*C Quaresma et al., 2016*; *Fujinaga, 2020*). Dysregulation of the P-TEFb equilibrium contributes to various diseases such as solid tumors (mutation in LARP7), leukemias and lymphomas (DNA translocations leading to aberrant recruitment of P-TEFb), and cardiac hypertrophy (inactivation of HEXIM1) (*Franco et al., 2018*). Cellular stresses such as ultraviolet (UV) irradiation and heat, as well as various small compounds such as histone deacetylase inhibitors (HDACi), HMBA and bromodomain extra-terminal domain (BET) inhibitors (JQ1), also promote the release of P-TEFb from the 7SK snRNP in a reversible manner (*Zhou et al., 2012*). Despite these important cellular responses, different viruses, such as HIV, HTLV, EBV, HSV, HCMV, and others, have evolved different strategies to utilize P-TEFb for their own replication (*Mbonye et al., 2013*; *Zaborowska et al., 2016*). For example, the HIV transactivator of transcription (Tat) not only binds to free P-TEFb but also promotes the release of P-TEFb from 7SK snRNP. Tat:P-TEFb then binds to the trans-activation response (TAR) RNA stem loop to activate the transcription of viral genes (*Selby and Peterlin, 1990*; *Wei et al., 1998*). Additionally, the establishment of viral latency in quiescent cells parallels the disappearance of P-TEFb, which can be reversed by cell activation (*Rice, 2019*).

In activated and proliferating cells, high levels of P-TEFb are found. In contrast, they are vanishingly low in resting cells, especially monocytes and memory T cells. While levels of CDK9 persist, those of CycT1 are greatly reduced. At the same time, transcripts for CycT1 and CDK9 remain high in all these cells (*Garriga et al., 1998*; *Ghose et al., 2001*). Based on existing studies, one can conclude that P-TEFb falls apart when cells become quiescent. In these cells, CDK9 is stabilized by chaperone proteins HSP70 and HSP90 (*O'Keeffe et al., 2000*). For CycT1, it was thought that its translation is inhibited by RNAi (*Chiang and Rice, 2012*; *Sung and Rice, 2009*). In contrast, we find that CycT1 is rapidly degraded in these cells. We also identified post-translational modifications that lead to the assembly and disassembly of P-TEFb, which involves specific kinases and phosphatases. Importantly, the unbound CycT1 protein can be stabilized by proteasomal inhibitors. This situation appears uncannily reminiscent of cell cycle cyclins/CDKs that are also regulated by similar post-translational mechanisms.

## Results

### Critical residues in CycT1 (Thr143 and Thr149) are required for its binding to CDK9

Previously, residues in the N-terminal region of CycT1 (positions 1–280, CycT1(280), including cyclin boxes, positions 30–248) (*Figure 1A*) were found to be required for interactions between CycT1 and CDK9 (*Garber et al., 1998*). In particular, a substitution of the leucine to proline at position 203 (L203P) or four substitutions from a glutamic to aspartic acid at position 137 and threonine to alanine at positions 143, 149, and 155 (4MUT) completely abolished this binding (*Kuzmina et al., 2014*; *Figure 1A*). We created a similar set of mutant CycT1 proteins in the context of the full-length CycT1 and truncated CycT1(280) proteins (*Figure 1A*). Next, we defined further critical residues involved in CycT1:CDK9 interactions, especially the three adjacent threonine residues in the cyclin box (CycT1T3A, *Figure 1A*). First, these mutant CycT1 proteins were expressed in 293T cells. Next, interactions between mutant CycT1 proteins and the endogenous CDK9 protein were analyzed by co-immunoprecipitation (co-IP) (*Figure 1B–D*).

Mutant CycT1 proteins were poorly expressed in 293T cells (*Figure 1B*, panel 1, lanes 3, 5, and 7). Expression levels of these proteins were restored by incubating cells with the potent and clinically approved proteasomal inhibitor bortezomib (*Figure 1B*, panel 1, compare lanes 4, 6, and 8 to lanes 3, 5, and 7), indicating that these mutant CycT1 proteins are highly unstable in cells. To confirm this finding, the half-life of mutant CycT1 proteins was measured by cycloheximide (CHX) pulse-chase experiments (*Figure 1C*). Whereas levels of the wild-type (WT) CycT1 protein remained unchanged after CHX treatment (*Figure 1C*, panel 1, lanes 1–4), those of three mutant CycT1 proteins (CycT1L203P, CycT14MUT, and CycT1T3A) decreased rapidly with the half-life of ~3 hr, ~ 2.5 hr, and ~6 hr, respectively (*Figure 1C*, panel 1, lanes 5–8; lanes 9–12; lanes 13–16). Moreover, levels of the endogenous CDK9 protein were not changed under bortezomib (*Figure 1B*, panel 2) or CHX treatment (data not provided). When protein levels were restored by bortezomib, mutant CycT1 proteins (CycT1L203P and CycT14MUT) did not interact with CDK9 (*Figure 1D*, panel 1, compare lanes 4 and 5 to lane 3). Similarly, interactions between mutant CycT1T3A and CDK9 proteins were significantly decreased (*Figure 1D*, panel 1, compare lane 6 to lane 3, ~7.8-fold reduction).

We further examined whether all these threonine residues are important for binding to CDK9 (*Figure 1E*). Thus, mutant CycT1 proteins with single threonine substitution were created and examined for CycT1:CDK9 interactions. As presented in *Figure 1E*, whereas mutant CycT1T143A and CycT1T149A proteins exhibited significantly impaired binding to CDK9, the mutant CycT1T155A protein did not (panel 1, compare lanes 4 and 5 to lanes 2 and 6, ~5.2-fold and ~3.7-fold reduction). These results indicate that Thr143 and Thr149, but not Thr155, are critical residues in CycT1 for binding to CDK9. Finally, the mutant CycT1(280) protein with double threonine substitution to alanine (CycT1(280)TT143149AA) demonstrated reduced binding to CDK9 to a similar extent as the mutant CycT1(280)T3A protein (*Figure 1F*, panel 1, compare lanes 4 and 5 to lane 2, ~7.2-fold and ~8.1-fold reduction). Taken together, we conclude that two threonine residues (Thr143 and Thr149) in CycT1 are critical for its binding to CDK9 to form P-TEFb, and that mutant CycT1 proteins that do not interact or poorly interact with CDK9 are rapidly degraded by the proteasome.

### Phosphorylation of Thr143 and Thr149 in CycT1 contributes to its binding to CDK9

Since threonine residues are potential phosphorylation sites, we examined whether phosphorylation of Thr143 and/or Thr149 in CycT1 contributes to P-TEFb assembly. 293T cells ectopically expressing CycT1 (*Figure 2A*) or CycT1(280) (*Figure 2B*) were treated with the potent protein phosphatase inhibitors okadaic acid for 1.5 hr or calyculin A for 1 hr prior to cell lysis. Following the co-IP of CDK9, its phosphorylation was analyzed with anti-phospho-threonine (pThr) antibodies by western blotting (WB). Phospho-threonine signals were increased in CycT1 and CycT1(280) proteins in the presence of high concentrations (1 µM) of okadaic acid and 150 nM calyculin A, which inhibit serine/threonine protein phosphatases (PP) 1 and 2A (*Figure 2A and B*, panel 3, lanes 4 and 5), but not by a low concentration (5 nM) of okadaic acid (*Figure 2A and B*, panel 3, lane 3), which only inhibits PP2A. Confirming these inhibitory effects by okadaic acid and calyculin A, CycT1 and CDK9 bands shifted upward by this treatment, and only these upper bands were detected with anti-pThr antibodies

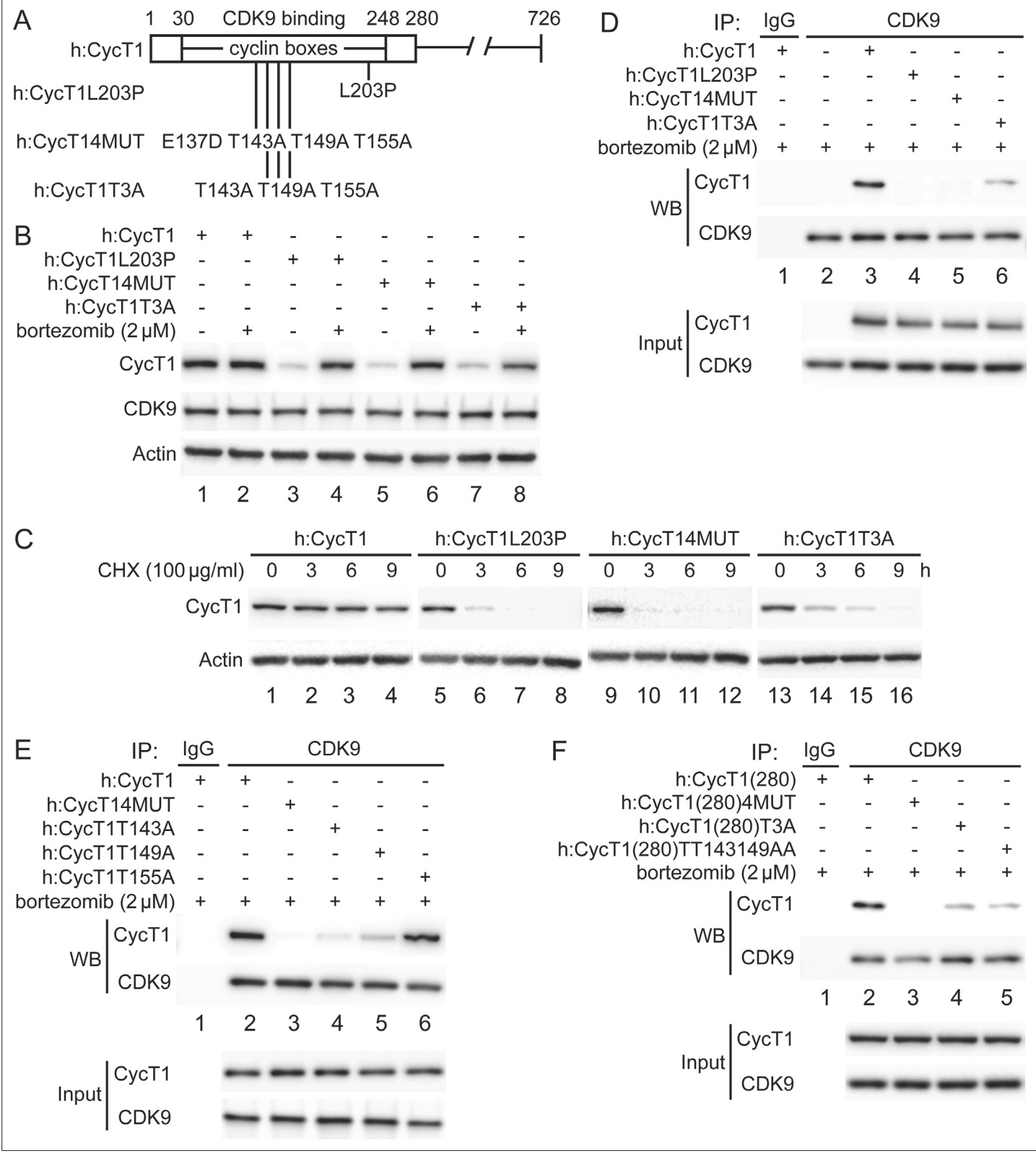

**Figure 1.** Critical residues in CycT1 (Thr143 and Thr149) are required for its binding to CDK9. (**A**) Diagram of WT CycT1 and indicated mutant CycT1 proteins. The full-length human CycT1 protein contains 726 residues. Two cyclin boxes are found between positions 30 and 248. Critical residues for CDK9 binding include Thr143, Thr149, and Thr155. Glu137 and Leu203 flank these sites. Presented are critical mutations in CycT1 that form the basis of this study. (**B**) Mutant CycT1 proteins are unstable. WT CycT1 and three indicated mutant CycT1 proteins were expressed in 293T cells, which were

*Figure 1 continued on next page*

*Figure 1 continued*

untreated (lanes 1, 3, 5, and 7) or treated with 2 μM bortezomib for 12 hr (lanes 2, 4, 6, and 8) before cell lysis. Levels of CycT1 (panel 1), CDK9 (panel 2), and the loading control actin (panel 3) proteins were detected with anti-HA, anti-CDK9, and anti-β-actin antibodies, respectively, by WB. Top panels are designated as panel 1, and panel numbers increase from top to bottom (same numbering rules are applied to all WB panels throughout). Gels are marked as follows: IP, IPed proteins, above panels; next, presence and absence of co-IPed proteins is denoted by (+) and (-) signs; same for the inclusion and concentration of bortezomib; WB, western blot of co-IPed proteins; Input, western blot of input proteins. (**C**) Mutant CycT1 proteins are unstable. WT CycT1 and three indicated mutant CycT1 proteins were expressed in 293T cells, which were untreated (lanes 1, 5, 9, and 13) or treated with 100 μg/ml cycloheximide (CHX) for 3–9 hr (lanes 2–4; 6–8; 10–12; 14–16) before cell lysis. Levels of CycT1 (panel 1) and the loading control actin (panel 2) proteins were detected with anti-HA and anti-β-actin antibodies, respectively, by WB. (**D**) Interactions between CDK9 and mutant CycT1 proteins are impaired. WT CycT1 and three indicated mutant CycT1 proteins were expressed in 293T cells treated with bortezomib. Co-IPs with CDK9 are presented in panels 1 and 2. Panels 3 and 4 contain input levels of CycT1 and CDK9 proteins. (**E**) Interactions between CDK9 and point mutant CycT1 proteins are impaired. WT CycT1 and four indicated mutant CycT1 proteins were expressed in 293T cells treated with bortezomib. Co-IPs with CDK9 are presented in panels 1 and 2. Panels 3 and 4 contain input levels of CycT1 and CDK9 proteins. (**F**) Interactions between CDK9 and truncated mutant CycT1(280) proteins are impaired. WT CycT1 and three indicated mutant CycT1 proteins were expressed in 293T cells treated with bortezomib. Co-IPs with CDK9 are presented in panels 1 and 2. Panels 3 and 4 contain input levels of CycT1(280) and CDK9 proteins.

(*Figure 2*). Furthermore, interactions between CycT1 and CDK9 were increased by high concentrations of okadaic acid and calyculin A (*Figure 2A*, panel 1, compare lane 4 to lane 2 and lane 5 to lane 2, ~6.1-fold and ~4.2-fold increase). In addition, CycT1 co-IPed with CDK9 was heavily phosphorylated at threonine residues (*Figure 2A*, panel 3). Under the same conditions, CDK9 was also heavily phosphorylated (*Figure 2A*, panel 4). Similarly, interactions between CycT1(280) and CDK9 were also increased by high concentrations of okadaic acid and calyculin A (*Figure 2B*, panel 1, compare lane 4 to lane 2 and lane 5 to lane 2, ~5.1-fold and ~4.7-fold increase). Significant threonine phosphorylation of CDK9-associated CycT1(280) and CDK9 was also detected (*Figure 2B*, panels 3 and 4), although increased phosphorylation of CDK9 did not correlate with increased interactions with CycT1 (*Figure 2A and B*, compare lane 5 to lane 4).

An online database for phosphorylation site prediction (NetPhos 3.1, developed by Technical University of Denmark) scores Thr143 and Thr149 above the threshold value (default 0.5), indicating that these threonines are potential phosphorylation sites (*Figure 2—figure supplement 1A*). To examine whether Thr143 and Thr149 are phosphorylated in CycT1, levels of total threonine phosphorylation were compared between WT CycT1(280) and mutant CycT1(280)TT143149AA proteins in the presence of 1 μM okadaic acid. As presented in *Figure 2C*, levels of threonine phosphorylation were significantly reduced in the mutant CycT1(280)TT143149AA protein compared to WT CycT1(280) (panel 1, compare lane 3 to lane 2, ~4.7-fold reduction), indicating that Thr143 and Thr149 are phosphorylated within CycT1(280). Of note, the mutant CycT1(280)TT143149AA protein migrated to a similar extent as WT CycT1(280) in the presence of okadaic acid, which implies that there are additional phosphorylation sites in CycT1 besides these two threonines. Indeed, WBs with anti-phospho-serine and anti-phospho-tyrosine antibodies confirmed that WT CycT1(280) and the mutant CycT1(280)TT143149AA proteins were phosphorylated on these additional residues to similar levels (*Figure 2—figure supplement 1B*).

To further demonstrate that these two threonines in CycT1 are phosphorylated in vivo, we performed phospho-peptide mapping analyses by the validated in-gel phospho-staining with a specific phosphoprotein dye (Phospho-Tag). For the purpose of tryptic peptide mapping, we chose a further truncated CycT1 protein to position 192 (CycT1(192)). In contrast to two predicted tryptic fragments of equivalent size for CycT1(280) protein, only one large tryptic peptide with a predicted mass 4.3 kD (a.a. 123–159, containing Thr143 and Thr149 residues) is found in the CycT1(192) protein, all others being of much smaller size. These predictions were generated by the Expasy website. Next, WT CycT1(192) and mutant CycT1(192)TT143149AA proteins were expressed in 293T cells with or without okadaic acid, and purified by IP. WB assay of the IPed proteins gave similar band shift as WT CycT1(280) and its mutant proteins (compare *Figure 2—figure supplement 1C* to *Figure 2C*). Indeed, as presented in *Figure 2D*, after validation of the specificity by in-gel Phospho-Tag staining, levels of phosphorylation in the mutant CycT1(192)TT143149AA protein were also much lower than those in the WT CycT1(192) in the presence of 1 μM okadaic acid (*Figure 2D*, compare lane 3 to lane 2, ~4.1-fold decrease). No phosphorylation signals were detected in the absence of okadaic acid (lane 1). The unphosphorylated BSA protein served as the negative control (lane 4).

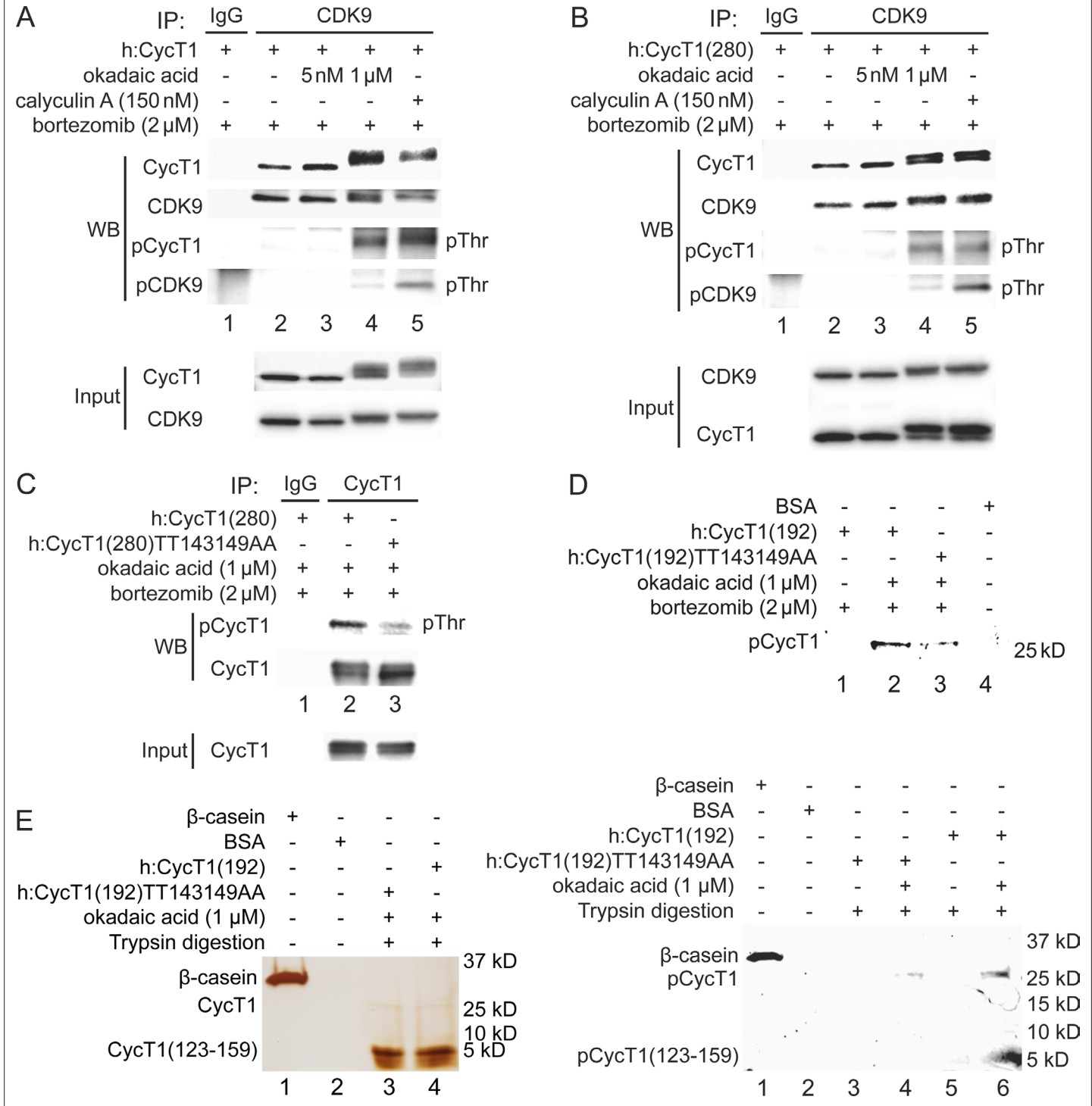

**Figure 2.** Phosphorylation of Thr143 and Thr149 in CycT1 contributes to its binding to CDK9. (**A**) Threonine phosphorylation is detected in the full-length CycT1 protein. CycT1 was expressed in 293T cells untreated or treated with 5 nM or 1 µM okadaic acid, or 150 nM calyculin A (+/- signs on top). Co-IPs with CDK9 were then probed with anti-HA and anti-CDK9 antibodies in panels 1 and 2, with anti-phospho-threonine (pThr) antibodies in panels 3 and 4. Panels 5 and 6 contain input levels of CycT1 and CDK9 proteins. (**B**) Threonine phosphorylation is detected in CycT1(280). CycT1(280) protein was expressed in 293T cells untreated or treated with 5 nM or 1 µM okadaic acid, or 150 nM calyculin A (+/- signs on top). Co-IPs with CDK9 were then probed with anti-HA and anti-CDK9 antibodies in panels 1 and 2, with anti-pThr antibodies in panels 3 and 4. Panels 5 and 6 contain input levels of CycT1 and CDK9 proteins. (**C**) Thr143 and Thr149 are major phospho-threonine residues in CycT1(280). WT CycT1(280) or mutant CycT1(280) TT143149AA proteins were expressed in the presence of bortezomib and 1 µM okadaic acid in 293T cells. IPs with CycT1 were then probed with anti-

*Figure 2 continued on next page*

*Figure 2 continued*

pThr and anti-HA antibodies in panels 1 and 2. Panel 3 contains input levels of CycT1 proteins. (**D**) Thr143 and Thr149 are major phosphorylated residues in CycT1(192). WT CycT1(192) or mutant CycT1(192)TT143149AA proteins were expressed in the presence of bortezomib and/or 1 µM okadaic acid (+/- signs on top) in 293T cells. After IPs with anti-HA antibodies, IPed samples were subjected to SDS-PAGE, then phosphorylated proteins were detected by in-gel Phospho-Tag staining, with unphosphorylated BSA protein as the negative control. (**E**) Direct detection of Thr143/Thr149 phosphorylation by phosphopeptide mapping analysis. WT CycT1(192) or mutant CycT1(192)TT143149AA proteins were expressed in the presence of bortezomib and/or 1 µM okadaic acid (+/- signs on top) in 293T cells. After IP with anti-HA antibodies. IPed samples were digested by trypsin and subjected to SDS-PAGE, followed by silver staining (left panel) in-gel Phospho-Tag staining (right panel), using phosphorylated β-casein protein as the positive control and unphosphorylated BSA protein as the negative control.

The online version of this article includes the following figure supplement(s) for figure 2:

**Figure supplement 1.** Thr143 and Thr149 are main phosphorylation sites in CycT1.

Finally, purified WT CycT1(192) and mutant CycT1(192)TT143149AA proteins were subjected to trypsin digestion, followed by separation by 4–20% SDS-PAGE. Tryptic peptides containing phosphorylated residues were detected by silver and in-gel Phospho-Tag staining. As presented in *Figure 2E*, actual sizes of tryptic peptides of WT CycT1(192) and mutant CycT1(192)TT143149AA proteins were confirmed by silver-stained PAGE (right panel, lanes 3 and 4, same as the prediction). Next, the in-gel Phospho-Tag staining of tryptic peptides of WT CycT1(192) detected a phosphorylated peptide with the same size as the top band in silver-stained PAGE (*Figure 2E*, right panel, lane 6). Importantly, no phosphorylated peptide of the corresponding size was detected with the same peptide from the mutant CycT1(192)TT143149AA protein (*Figure 2E*, right panel, lane 4) or undigested samples (*Figure 2E*, right panel, lanes 3 and 5). The phosphorylated β-casein served as the positive control (*Figure 2E*, right panel, lane 1) and unphosphorylated BSA served as the negative control (*Figure 2E*, right panel, lane 2).Thus, the phosphorylation of Thr143 and Thr149 was detected by two independent methods, WB with anti-phospho-threonine antibodies and direct Phospho-Tag staining of tryptic peptides. Taken together, these findings strongly indicate that CycT1 is phosphorylated at Thr143 and Thr149, which potentiates its binding to CDK9, and that PP1 is involved in the dephosphorylation of CycT1.

## Residues in CycT1 and CDK9 that regulate the assembly of P-TEFb

Thr143 and Thr149 are located in the region of CycT1 that binds to CDK9. Spatial locations of these threonines are based on two published crystal structures of the human P-TEFb complex (PDB access code 3MI9) (*Baumli et al., 2008*; *Tahirov et al., 2010*). Since P-TEFb was expressed and purified from insect cells, CycT1 was most likely already phosphorylated. Importantly, no one has been able to assemble P-TEFb from prokaryotic cells, such as *Escherichia coli* (*Baumli et al., 2008*; *Schulze-Gahmen et al., 2014*; *Schulze-Gahmen et al., 2013*; *Tahirov et al., 2010*). To understand the role of CycT1 Thr143 and Thr149 phosphorylation in P-TEFb, the model was created by adding phosphates to Thr143 and Thr149 in the published crystal structure of P-TEFb (PDB ID 3MI9), followed by energy minimization and molecular dynamics (MD) simulations (*Figure 3A*). As presented in *Figure 3A*, the side chain of Thr143 is placed internally towards helices in the cyclin box 1 of CycT1, where Gln73 is its nearest contact residue, suggesting that Thr143 is involved in intramolecular interactions with Gln73. On the other hand, Thr149 is located in the interface between CycT1 and CDK9 where Lys68 in CDK9 is its nearest contact residue, suggesting that Thr149 is involved in intermolecular interactions between CycT1 and CDK9. The space between CycT1's Gln73 and Thr143, and between CycT1's Thr149 and CDK9's Lys68 readily accommodates phosphate molecules on Thr143 and Thr149 in CycT1. Indeed, the predominant contribution to the increased binding energy of the complex comes from electrostatic ($\Delta E_{elec}$) and polar solvation ($\Delta E_{solv-polar}$) energies, which is consistent with the stabilizing interactions described in previous figures. Further, we performed molecular mechanics/generalized born surface area calculations (MM-GBSA) (*Pettersen et al., 2004*), which predicted that the phosphorylation of Thr143 and Thr149 in CycT1 can be thermodynamically advantageous for interactions between CDK9 and CycT1 (*Table 1*). We then tested these predictions experimentally.

WT CDK9 or mutant CDK9K68A proteins were coexpressed with WT CycT1(280) or mutant CycT1(280)Q73A proteins in the presence of bortezomib for 12 hr before co-IP. As presented in *Figure 3B*, compared to WT CycT1(280), the mutant CycT1(280)Q73A protein exhibited a lower affinity for CDK9 (panel 1, compare lane 3 to lane 2, ~5.3-fold reduction). Similarly, compared to WT

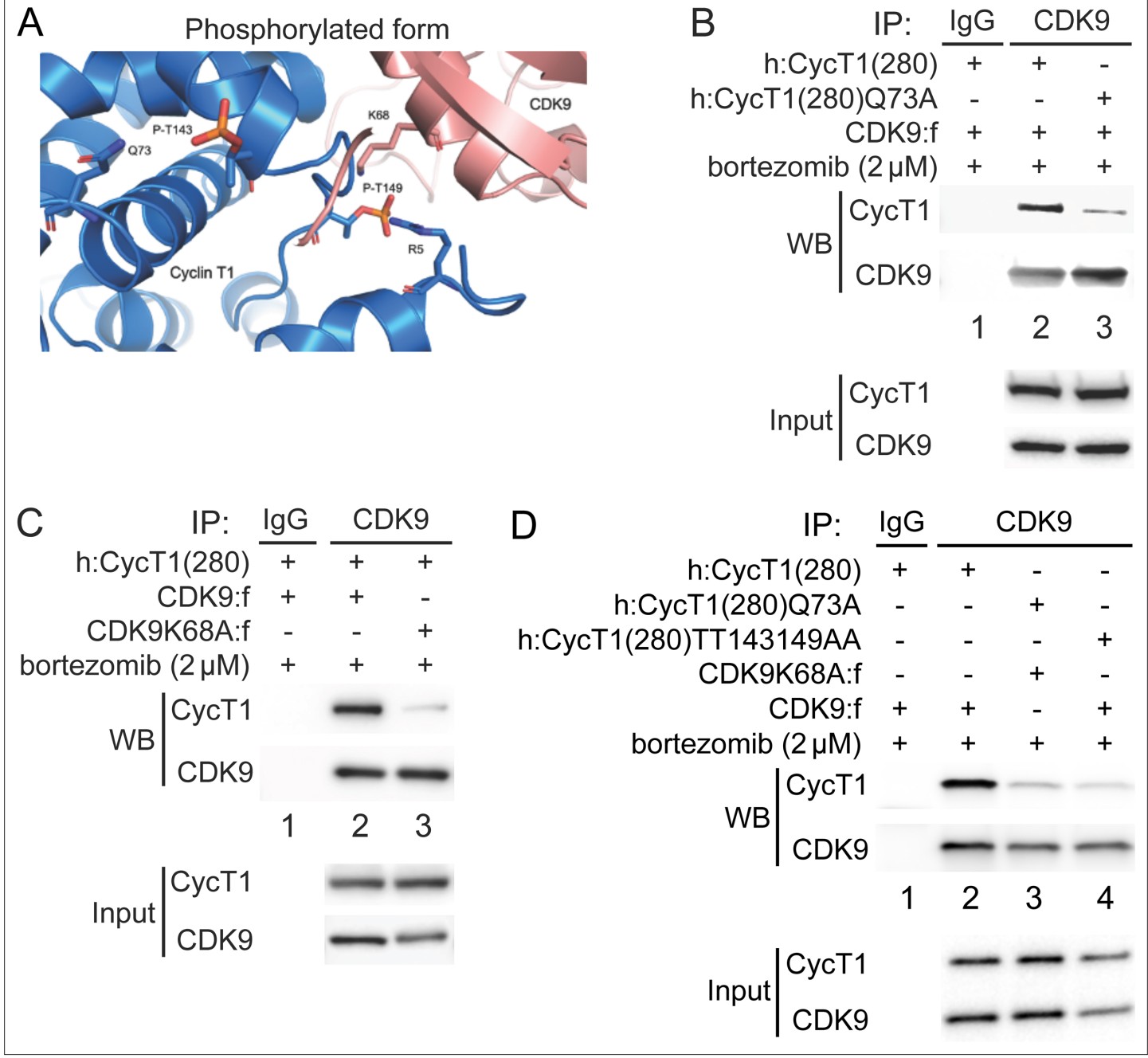

**Figure 3.** Phosphorylation of Thr143 and Thr149 stabilizes the interface between CycT1 and CDK9. (**A**) Model of P-TEFb where CycT1 is phosphorylated at Thr143 and Thr149. The model was created by adding phosphates to Thr143 and Thr149 in the published crystal structure of P-TEFb (PDB ID 3MI9), followed by energy minimization and molecular dynamics (MD) simulations. Residues predicted to interact with Thr143 and Thr149 are Gln73 in CycT1 and Lys68 in CDK9, respectively. (**B**) Gln73 is targeted by phosphorylated Thr143 in CycT1. WT CycT1(280) or mutant CycT1(280)Q73A proteins and CDK9 were coexpressed in the presence of bortezomib (+/- signs on top) in 293T cells. Co-IPs with CDK9 are presented in panels 1 and 2. Panels 3 and 4 contain input levels of CycT1 and CDK9 proteins. (**C**) Lys68 in CDK9 is targeted by phosphorylated Thr149 in CycT1. WT CDK9 or mutant CDK9K68A proteins and CycT1(280) were coexpressed in the presence of bortezomib (+/- signs on top) in 293T cells. Co-IPs with CDK9 are presented in panels 1 and 2. Panels 3 and 4 contain input levels of CycT1 and CDK9 proteins. (**D**) Mutations of K68A in CDK9 and Q73A in CycT1 attenuate cooperatively the binding between CycT1 and CDK9, equivalently to the mutant CycT1TT143149AA protein. WT CycT1(280) or mutant CycT1(280)Q73A proteins and WT CDK9 or mutant CDK9K68A proteins were coexpressed in the presence of bortezomib (+/- signs on top) in 293T cells. Co-IPs with CDK9 are presented in panels 1 and 2. Panels 3 and 4 contain input levels of CycT1 and CDK9 proteins.

**Table 1.** Summary of binding energies calculated using molecular mechanics/generalized born surface area (MM-GBSA) calculations.

The final binding energies ($\Delta G_{bind}$) are shown in bold. $\Delta E_{elec}$, $\Delta E_{wdV}$, $\Delta E_{GB}$, and $\Delta E_{surf}$ correspond to the electrostatic energy, van der Waals energy, polar solvation energy, and non-polar solvation energy contributions, respectively. Standard deviations of the mean are shown in parenthesis.

| Contribution | WT (kcal/mol) | PThr$_{143}$ (kcal/mol) | PThr$_{149}$ (kcal/mol) | PThr$_{143,149}$ (kcal/mol) |
|---|---|---|---|---|
| $\Delta G_{bind}$ | **−82.1** (11.4) | **−85.0** (13.8) | **−81.6** (12.8) | **−98.5** (14.3) |
| $\Delta E_{elec}$ | −480.8 (62.6) | −849.0 (98.1) | −863.1 (107.9) | −1249.1 (86.1) |
| $\Delta E_{wdV}$ | −144.1 (10.8) | −158.1 (12.1) | −151.0 (12.4) | −142.6 (11.4) |
| $\Delta E_{GB}$ | 562.9 (62.4) | 944.8 (99.0) | 954.2 (105.5) | 1315.3 (83.5) |
| $\Delta E_{surf}$ | −20.1 (1.6) | −22.7 (1.7) | −21.7 (1.6) | −22.1 (1.7) |

CDK9, the mutant CDK9K68A protein exhibited a lower affinity for WT CycT1(280) (*Figure 3C*, panel 1, compare lane 3 to lane 2, ~5.9-fold reduction). Finally, interactions between mutant CycT1(280) Q73A and CDK9K68A proteins were reduced to a similar extent to those between the mutant CycT1(280)TT143149AA and WT CDK9 proteins, compared to the positive control with WT CDK9 and WT CycT1(280) (*Figure 3D*, panel 1, compare lanes 3 and 4 to lane 2, ~7.5-fold and ~8-fold reduction). Taken together, phosphates on Thr143 and Thr149 in CycT1 are essential for the assembly and stability of P-TEFb.

## PKC inhibitors impair interactions between CycT1 and CDK9 and promote CycT1 degradation

In resting and memory T cells, levels of CycT1 are vanishingly low. In previous sections, we discovered that Thr143 and Thr149 are phosphorylated in the stable P-TEFb complex. Using kinase prediction programs (NetPhos 3.1), these residues lie in separate PKC consensus sites. While Thr143 received the highest score, Thr149 could also be a target for PKC or other kinases. These findings implied that PKC family members are kinases that phosphorylate Thr143 and/or Thr149.

To examine whether PKC promotes the phosphorylation and stability of CycT1, several PKC inhibitors were introduced to different cells (*Figure 4*). Of these, staurosporine exhibited the most significant inhibition in 293T cells. As presented in *Figure 1B and C*, the exogenous CycT1 protein is very stable in cells. Next, increasing amounts of staurosporine were added to cells 12 hr prior to cell lysis. Staurosporine reduced levels of CycT1 in these cells in a dose-dependent manner (*Figure 4—figure supplement 1A*, , panel 1, compare lanes 2 and 3 to lane 1, ~4-fold and ~16-fold reduction). In addition, interactions between the exogenous CycT1 and endogenous CDK9 proteins were inhibited by staurosporine even under conditions where the expression of CycT1 was restored by bortezomib (*Figure 4A*, panel 1, compare lane 3 to lane 2, ~8.1-fold reduction). Similarly, interactions between CycT1(280) and CDK9 were decreased by staurosporine and two additional, more specific PKC inhibitors bisindolylmaleimide IX and HBDDE (*Figure 4B*, compare lanes 4–6 to lane 2, ~ 8.7-fold, 7.1-fold, and 6.5-fold decrease). In contrast, the MEK 1/2 inhibitor (MEK 1/2i) had no effect (*Figure 4B*, compare lane 3 to lane 2).

To examine whether PKC inhibition antagonizes the phosphorylation of CycT1(280) in the presence of the high concentration of okadaic acid, cells expressing the exogenous CycT1(280) protein in the presence of bortezomib were treated with the same set of kinase inhibitors before adding okadaic acid for another 1.5 hr. Levels of threonine phosphorylation of CycT1(280) were compared by WB after IPs with anti-HA antibodies. As presented in *Figure 4C*, three PKC inhibitors, but not MEK 1/2i, all antagonized the threonine phosphorylation of CycT1(280) (panel 1, compare lanes 4–6 to lane 2, ~13.3-fold, ~8.3-fold, and ~5.7-fold decrease). Thus, PKC inhibitors, especially staurosporine, inhibit the phosphorylation of CycT1 and its interactions with CDK9, which results in the degradation of CycT1 in 293T cells.

To validate further the specificity of such negative regulation by PKC inhibitors, Jurkat and activated primary CD4+ T cells were also treated with different PKC inhibitors at increasing amounts for 12 hr. As presented in *Figure 4D* and compared to untreated cells, levels of endogenous CycT1

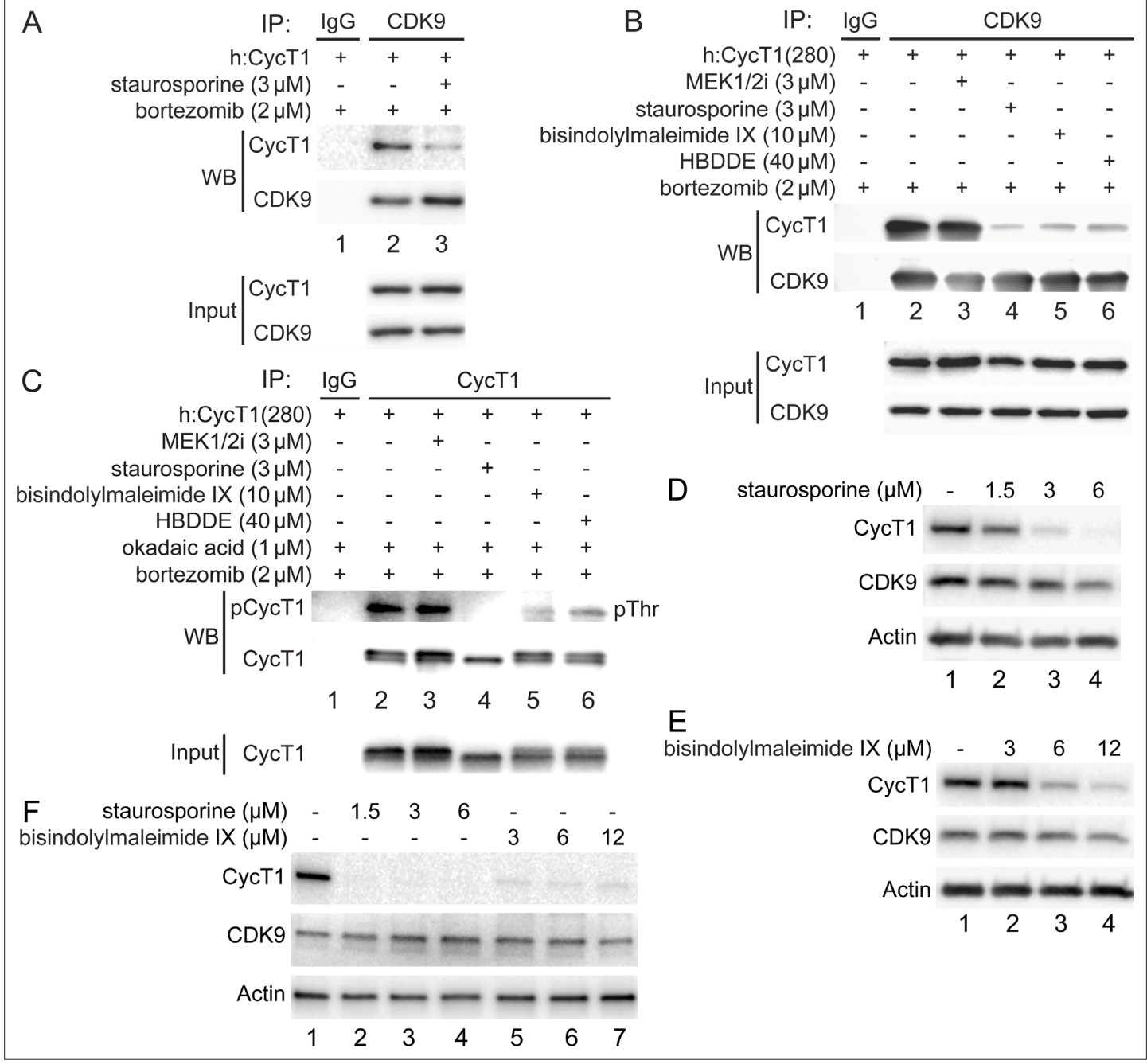

**Figure 4.** PKC inhibitors impair interactions between CycT1 and CDK9, and promote CycT1 degradation. (**A**) PKC inhibitors impair interactions between CycT1 and CDK9. CycT1 and CDK9 were coexpressed in the presence or absence of staurosporine and bortezomib (+/- signs on top) in 293T cells. Co-IPs with CDK9 are presented in panels 1 and 2. Panels 3 and 4 contain input levels of CycT1 and CDK9 proteins. (**B**) PKC inhibitors impair interactions between CycT1(1–280) and CDK9. CycT1(280) was expressed in the presence or absence of bortezomib and indicated concentration of MEK 1/2i, staurosporine, bisindolylmaleimide IX, or HBDDE (+/- signs on top) in 293T cells. Co-IPs with CDK9 are presented in panels 1 and 2. Panels 3 and 4 contain input levels of CycT1(280) and CDK9 proteins. (**C**) PKC inhibitors inhibit threonine phosphorylation of CycT1. CycT1(280) was expressed in the presence or absence of bortezomib and indicated concentration of MEK 1/2i, staurosporine, bisindolylmaleimide IX, or HBDDE (+/- signs on top) in 293T cells. IPs with CycT1 are presented in panels 1 and 2. Phosphorylated proteins were visualized with anti-pThr antibodies (panel 1). Panel 3 contains input levels of CycT1(280) proteins. (**D**) Staurosporine decreases CycT1 levels in a dose-dependent manner. Jurkat cells were untreated (lane 1) or treated with increasing doses of staurosporine (lanes 2–4) for 12 hr before cell lysis. Levels of CycT1 (panel 1), CDK9 (panel 2), and the loading control actin (panel 3) proteins were detected with anti-CycT1, anti-CDK9, and anti-β-actin antibodies, respectively, by western blotting (WB). (**E**) PKC inhibitor bisindolylmaleimide IX decreases CycT1 levels in a dose-dependent manner. Jurkat cells were untreated (lane 1) or treated with increasing doses of bisindolylmaleimide IX (lanes 2–4) for 12 hr before cell lysis. Levels of CycT1 (panel 1), CDK9 (panel 2), and the loading control actin (panel 3) proteins

*Figure 4 continued on next page*

Figure 4 continued

were detected with anti-CycT1, anti-CDK9, and anti-β-actin antibodies, respectively, by WB. (F) CycT1 levels in activated primary CD4+ T cells are decreased by PKC inhibitors in a dose-dependent manner. Activated primary CD4+ T cells were untreated (lane 1) or treated with increasing amounts of staurosporine (lanes 2–4), bisindolylmaleimide IX (lanes 5–7) for 12 hr before cell lysis. Levels of CycT1 (panel 1), CDK9 (panel 2), and the loading control actin (panel 3) were detected with anti-CycT1, anti-CDK9, and anti-β-actin antibodies, respectively, by WB.

The online version of this article includes the following figure supplement(s) for figure 4:

**Figure supplement 1.** PKC inhibitors promote CycT1 degradation in different cells.

protein in Jurkat cells were significantly decreased by staurosporine in a dose-dependent manner (panel 1, compare lanes 2–4 to lane 1: 1.5 μM, ~2-fold; 3 μM, ~11-fold; 6 μM, ~20-fold reduction). Levels of CDK9 were largely unaffected (panel 2 in *Figure 4D*). Similar to staurosporine, two additional PKC inhibitors, bisindolylmaleimide IX (*Figure 4E*) and H-7 (*Figure 4—figure supplement 1B*), also decreased levels of the endogenous CycT1 protein in a dose-dependent manner. As presented in *Figure 4E*, levels of CycT1 were decreased by bisindolylmaleimide IX ~4-fold at 6 μM and ~10-fold at 12 μM (panel 1, compare lanes 3 and 4 to lane 1). PKC inhibitor H-7 also decreased levels of CycT1 ~6-fold at 70 μM and ~13-fold at 100 μM (*Figure 4—figure supplement 1B*, panel 1, compare lanes 3 and 4 to lane 1). Levels of CDK9 were also largely unaffected by these inhibitors (*Figure 4D*, *Figure 4—figure supplement 1B*, panels 2).

Staurosporine and bisindolylmaleimide IX were also used in activated primary CD4+ T cells to confirm further our findings (*Figure 4F*). In these cells, staurosporine depleted CycT1 up to 30-fold at 1.5, 3, and 6 μM (donor 1, *Figure 4F*, panel 1, compare lanes 2–4 to lane 1). Bisindolylmaleimide IX decreased them up to 20-fold at 3, 6, and 12 μM (*Figure 4F*, panel 1, compare lanes 5–7 to lane 1). In cells from donor 2, these compounds had similar effects on levels of CycT1 (*Figure 4—figure supplement 1C*, , panel 1). Levels of CDK9 were largely unaffected by these inhibitors (*Figure 4F*, *Figure 4—figure supplement 1C*, panel 2). Activated primary CD4+ T cells were also treated with three additional PKC inhibitors (sotrastaurin, H-7, and HBDDE) for 12 hr. As presented in *Figure 4—figure supplement 1D*, all these PKC inhibitors decreased levels of CycT1 in a dose-dependent manner (panel 1) without affecting those of CDK9 (panel 2). In sharp contrast, MEK 1/2i and three other PKC inhibitors (*Figure 4—figure supplement 1E*, compare lanes 2–6 to lane 1) had little effect on levels of CycT1 compared to the above four effective PKC inhibitors (*Figure 4—figure supplement 1E*, , compare lanes 7–10 to lane 1). Taken together, PKC inhibitors antagonize the phosphorylation of critical threonines in CycT1, which leads to the disassembly of P-TEFb and further degradation of CycT1.

## PKCα and PKCβ bind to CycT1, promote interactions between CycT1 and CDK9, and increase the stability of CycT1

Analysis of target specificities of our PKC inhibitors indicated that PKCα, PKCβ, PKCε represent candidate PKC isoforms responsible for the phosphorylation of CycT1. To validate that these PKC isoforms can target CycT1 for phosphorylation and promote P-TEFb assembly, their Flag-epitope-marked versions were expressed in 293T cells. Different dominant (kinase) negative mutant PKC isoforms were also coexpressed with WT CycT1(280) or mutant CycT1(280)TT143149AA proteins in the presence of bortezomib for 12 hr. Co-IPs were performed with anti-Flag antibodies. As presented in *Figure 5A*, the mutant PKCαK368R (*Soh and Weinstein, 2003*) protein interacted with the mutant CycT1(280)TT143149AA protein more potently than with WT CycT1(280) (panel 1, compare lane 4 to lane 3, approximately threefold increase), while no interactions with CDK9 were detected (*Figure 5A*, panel 2, lanes 3 and 4). As PKCβ1 and PKCβ2 only differ in their C-terminal 50 residues (*Kubo et al., 1987*), we employed the dominant negative PKCβ2K371R protein to block WT PKCβ protein. The dominant negative mutant PKCβ2K371R protein (*Soh and Weinstein, 2003*) was coexpressed with WT CycT1(280) or mutant CycT1(280)TT143149AA proteins. Co-IPs were performed with anti-Flag antibodies. As presented in *Figure 5B*, interactions between the mutant PKCβ2K371R protein and WT CycT1(280) or mutant CycT1(280)TT143149AA proteins were detected (panel 1, compare lane 4 to lane 3, ~4.1-fold increase). Again, CDK9 did not interact with the mutant PKCβ2K371R protein (*Figure 5B*, panel 2, lanes 3 and 4). In contrast, significantly reduced interactions were detected between WT CycT1(280) or mutant CycT1(280)TT143149AA proteins and PKCε, PKCδ, PKCγ, and

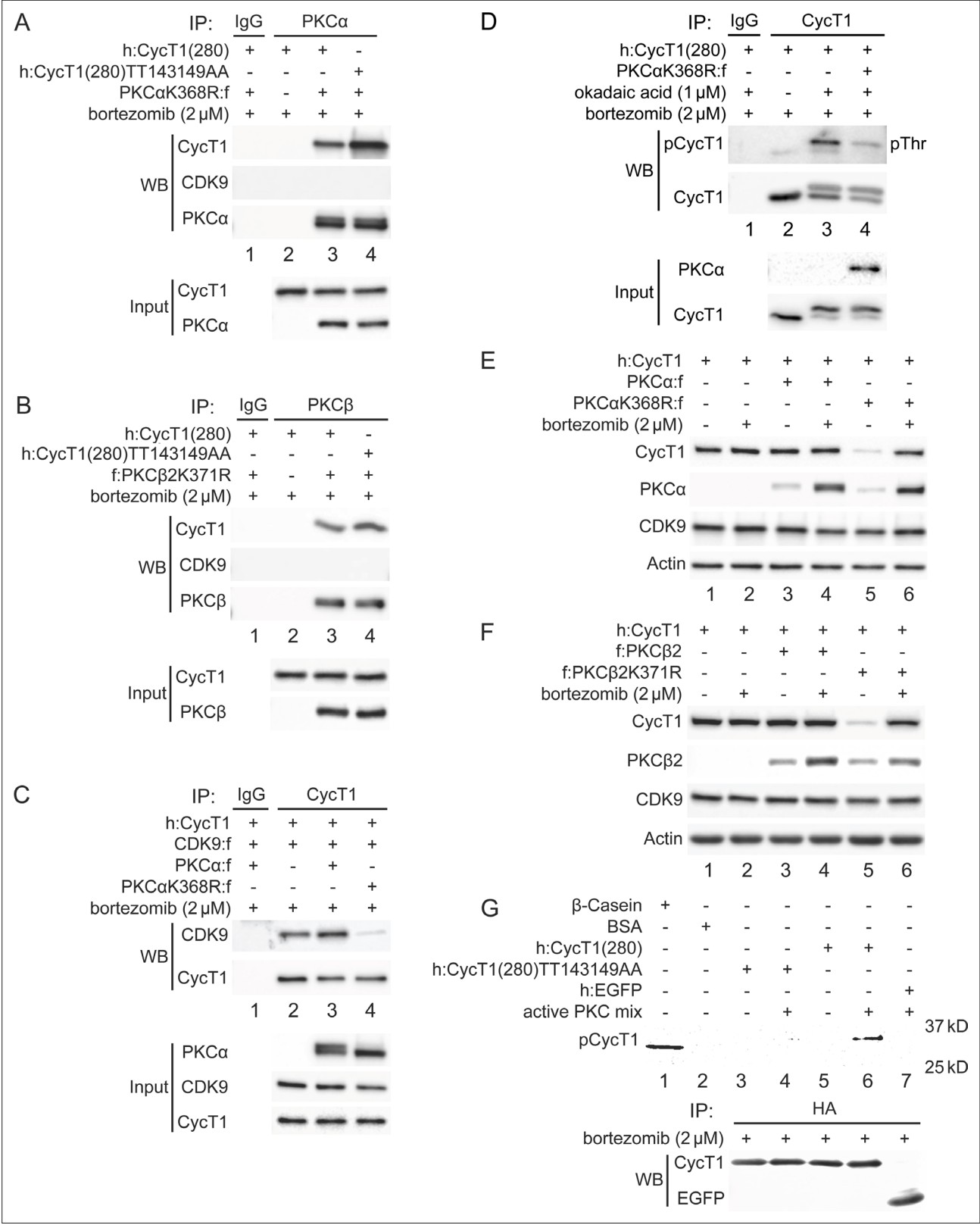

**Figure 5.** PKCα and PKCβ bind to CycT1 for its phosphorylation, promote interactions between CycT1 and CDK9, and increase the stability of CycT1. (**A**) PKCα binds to CycT1(280). Dominant negative mutant PKCαK368R and WT CycT1(280) or mutant CycT1(280)TT143149AA proteins were coexpressed in the presence of bortezomib (+/- signs on top) in 293T cells. Co-IPs with PKCα are presented in panels 1–3. Panels 4 and 5 contain input levels of CycT1(280) and PKCα proteins. (**B**) PKCβ binds to CycT1(280). Dominant negative mutant PKCβ2K371R and WT CycT1(280) or mutant CycT1(280)TT143149AA

*Figure 5 continued on next page*

*Figure 5 continued*

TT143149AA proteins were coexpressed in the presence of bortezomib (+/- signs on top) in 293T cells. Co-IPs with PKCβ are presented in panels 1–3. Panels 4 and 5 contain input levels of CycT1(280) and PKCβ proteins. (**C**) Dominant negative mutant PKCαK368R protein inhibits interactions between CDK9 and CycT1. PKCα or PKCαK368R, CycT1 and CDK9 were coexpressed in the presence of bortezomib (+/- signs on top) in 293T cells. Co-IPs with CycT1 are presented in panels 1 and 2. Panels 3–5 contain input levels of PKCα, CDK9, and CycT1 proteins. (**D**) PKCαK368R inhibits threonine phosphorylation of CycT1(280). CycT1(280) was expressed with or without PKCαK368R in the presence or absence of okadaic acid and bortezomib (+/- signs on top) in 293T cells. IPs with CycT1 were then probed with anti-pThr antibodies in panel 1 and with anti-HA antibodies in panel 2. Panels 3 and 4 contain input levels of PKCα and CycT1(280) proteins. (**E**) PKCαK368R decreases levels of CycT1 in cells. PKCα or PKCαK368R, and CycT1 were coexpressed in the presence or absence of bortezomib (+/- signs on top) in 293T cells. Levels of CycT1 (panel 1), PKCα (panel 2), CDK9 (panel 3), and the loading control actin (panel 4) were detected with anti-HA, anti-Flag, anti-CDK9, and anti-β-actin antibodies, respectively, by western blotting (WB). (**F**) PKCβ2K371R decreases levels of CycT1. PKCβ2 or PKCβ2K371R, and CycT1 were coexpressed in the presence or absence of bortezomib (+/- signs on top) in 293T cells. Levels of CycT1 (panel 1), PKCβ2 (panel 2), CDK9 (panel 3), and the loading control actin (panel 4) were detected with anti-HA, anti-Flag, anti-CDK9, and anti-β-actin antibodies, respectively, by WB. (**G**) PKCα and PKCβ phosphorylate Thr143 and Thr149 in CycT1 in vitro. WT CycT1(280) and mutant CycT1(280)TT143149AA proteins were expressed in the presence of bortezomib in 293T cells. After IP with anti-HA antibodies, equal levels of WT and mutant CycT1 proteins in IPed samples were detected by WB (panel 2). Immunoprecipitated proteins were incubated with purified constitutively active PKCα and PKCβ proteins for 2 hr and separated by SDS-PAGE, followed by in-gel Phospho-Tag staining. HA-tagged EGFP was used as the negative control for IP. Phosphorylated β-casein protein and unphosphorylated BSA protein represented positive and negative controls for in-gel Phospho-Tag staining, respectively.

The online version of this article includes the following figure supplement(s) for figure 5:

**Figure supplement 1.** PKCδ, PKCγ, and PKCε bind weakly to CycT1.

PKCθ (panel 1 in *Figure 5*; *Figure 5—figure supplement 1A and B* and 1C; data with PKCθ are not presented). We conclude that PKCα and PKCβ not only bind to but phosphorylate Thr143 and Thr149 in CycT1.

To confirm further that PKCα and PKCβ contribute to the assembly and stability of P-TEFb, WT PKCα or mutant PKCαK386R proteins were coexpressed with CycT1(280) and CDK9 in the presence of bortezomib (12 hr) in 293T cells. Interactions between these proteins were analyzed by co-IPs with anti-HA antibodies. As presented in *Figure 5C*, while PKCα slightly increased CycT1:CDK9 interactions (*Figure 5C*, panel 1, compare lane 3 to lane 2), the mutant PKCαK386R protein inhibited them (panel 1, compare lane 4 to lanes 2 and 3, ~7.9-fold reduction). To confirm that decreased interactions between CycT1(280) and CDK9 by the mutant PKCαK386R protein were caused by the inhibition of PKC-dependent phosphorylation of CycT1(280), it was coexpressed with the mutant PKCαK386R protein in the presence of bortezomib (12 hr) and okadaic acid (1.5 hr). IPs were conducted with anti-HA antibodies. As presented in *Figure 5D*, the expression of the mutant PKCαK386R protein decreased levels of threonine phosphorylation in CycT1(280) by ~5.2-fold as detected with anti-pThr antibodies (panel 1, compare lane 4 to lane 3). To demonstrate if the mutant PKCαK386R protein also decreased levels of CycT1 protein, WT PKCα or mutant PKCαK386R proteins were coexpressed with the CycT1 protein in the presence or absence of bortezomib (12 hr). As presented in *Figure 5E*, CycT1 coexpressed with PKCα had similar levels of expression as CycT1 without PKCα coexpression, which was not affected by bortezomib (panel 1, compare lane 3 to lane 1; compare lane 3 to lane 4; compare lane 1 to lane 2). In sharp contrast, coexpression of the mutant PKCαK386R protein decreased greatly levels of CycT1 protein in these cells (*Figure 5E*, panel 1, compare lane 5 to lanes 1 and 3, ~9.1-fold reduction), which was reversed by bortezomib (*Figure 5E*, panel 1, compare lane 6 to lane 5). Similar to the mutant PKCαK386R protein, the coexpressed mutant PKCβ2K371R protein also significantly diminished levels of CycT1 protein (*Figure 5F*, panel 1, compare lane 5 to lanes 1 and 3, ~10.4-fold reduction), which was reversed by bortezomib (*Figure 5F*, panel 1, compare lane 6 to lane 5). Levels of PKCs were increased by bortezomib, which is consistent with the demonstrated instability of PKC (*Lu et al., 1998*; *Figure 5E and F*, panels 2, compare lanes 3 and 5 to lanes 4 and 6). Also, levels of the endogenous CDK9 protein were unaffected in these cells (*Figure 5E and F*, panel 3).

Finally, we confirmed the phosphorylation of Thr143 and Thr149 of CycT1 by PKC in vitro. WT CycT1(280) and mutant CycT1(280)TT143149AA proteins were expressed in 293T cells and isolated by pull-down with HA-Ab-conjugated beads. They were incubated in the presence or absence of purified active PKC proteins (PKCα and PKCβ) in vitro. Phosphorylated proteins were separated by SDS-PAGE and detected by in-gel Phospho-Tag staining as in *Figure 2D and E*. As presented in *Figure 5G*, the WT CycT1(280), but not the mutant CycT1(280)TT143149AA protein, was detected by the in-gel Phospho-Tag staining in a manner that was dependent on the presence of PKC (panel 1, lanes 3–5,

and 7). Taken together, CycT1 is phosphorylated by PKCα and PKCβ, which not only promote interactions between CycT1 and CDK9, but also stabilize CycT1 in cells.

## Depletion of PKC leads to decreased levels of CycT1 in cells

Previous papers demonstrated that isoforms of PKC are inactive or absent in resting cells (*Heissmeyer et al., 2004*; *Pfeifhofer-Obermair et al., 2012*). Moreover, phorbol esters (PMA) deplete PKC in most cells (*Manger et al., 1987*). In addition, HIV Tat, whose proteome first identified and whose coactivator is P-TEFb, no longer works in these cells (*Jakobovits et al., 1990*). Together with our data, it appears that PKC influences dynamic changes of P-TEFb in different cell types and under varying conditions. To examine this situation further, 100 ng/ml PMA was administered to Jurkat cells or activated primary CD4+ T cells for several days. As presented in *Figure 6A*, endogenous CycT1 protein levels were decreased up to ~6-fold at 72 hr and ~11-fold at 96 hr after the addition of PMA (panel 1, compare lanes 2 and 3 to lane 1). Levels of CDK9 protein were largely unaffected (*Figure 6A*, panel 2, lanes 1–3). Furthermore, the same PMA treatment was performed in activated primary CD4+ T cells from two different donors. Similar to Jurkat cells, activated primary CD4+ T cells from donor 1 lost CycT1 expression up to ~7-fold at 72 hr and ~16-fold at 96 hr after PMA treatment (*Figure 6B*, panel 1, compare lanes 2 and 3 to lane 1). Again, levels of CDK9 were largely unaffected (*Figure 6B*, panel 2, lanes 1–3). Levels of PKCα were equivalently decreased at 72 and 96 hr after the addition of PMA (*Figure 6B*, panel 3, compare lanes 2 and 3 to lane 2). Other PKC isoforms, PKCβ1 and PKCβ2, were also depleted at these timepoints (*Figure 6B*, fourth and panel 5, compare lanes 2 and 3 to lane 2). These cells do not express PKCε (*Figure 6B*, panel 6, lanes 1–3). Same changes were also observed in activated primary CD4+ T cells from donor 2 (*Figure 6—figure supplement 1A*). We also found that the addition of bortezomib for another 24 hr after 72 hr PMA incubation rescued most of these decreased levels of CycT1 in activated primary CD4+ T cells (*Figure 6C*, panel 3, compare lane 3 to lane 2). Co-IPs with anti-CDK9 antibodies were also conducted in cells treated with bortezomib alone or with PMA and bortezomib. As presented in *Figure 6C*, interactions between CycT1 and CDK9 were significantly decreased in PMA-treated cells (up to ~7.6-fold) compared to controls (*Figure 6C*, panel 1, compare lane 3 to lane 2). These data demonstrate that the depletion of PKC in Jurkat and activated primary CD4+ T cells by PMA treatment causes the dissociation of P-TEFb and depletion of CycT1. This finding explains the hereto puzzling observation that Tat does not work in cells treated with PMA (*Jakobovits et al., 1990*).

We observed previously that levels of CycT1 increase significantly in resting CD4+ T cells with the addition of bortezomib (*Cary and Peterlin, 2020*). Nevertheless, interactions between CycT1 and CDK9 remain lower than in activated primary CD4+ T cells. To extend these findings to anergic T cells that lose the ability to respond to agonist antigen or stimulation of the T cell antigen receptor, we examined W131AOTII T cells from mice where the endogenous ZAP70 protein was substituted by a constitutively active mutant ZAP70-W131A protein. Introduction of this mutant protein into the OTII transgenic background (W131AOTII) results in high numbers of anergic and CD4 regulatory T cells (*Hsu et al., 2017*). As presented in *Figure 6D*, levels of CycT1 protein were significantly lower in W131AOTⅡ than in control OTⅡ T cells (panel 1, compare lane 2 to lane 1, ~7.8-fold decrease). Levels of the CDK9 were largely unchanged in these cells (*Figure 6D*, panel 2, compare lane 2 to lane 1). Meanwhile, levels of CycT1 and CDK9 transcripts in W131AOTⅡ and OTⅡ cells remained unchanged (*Figure 6E*), which is consistent with previous observations that mRNA levels of CycT1 and CDK9 do not vary between resting and activated CD4+ T cells (*Cary and Peterlin, 2020*; *Sung and Rice, 2006*). Moreover, since these W131AOTII T cells exhibit impaired T cell receptor signaling (*Nguyen et al., 2021*), activating these cells with anti-CD3 and anti-CD28 antibodies did not increase levels of CycT1 (data not presented).

It was demonstrated that treatment of T cells with calcium ionophores such as ionomycin also depletes PKC and induces anergy (*Heissmeyer et al., 2004*). Therefore, we examined whether sustained ionomycin treatment of primary activated CD4+ T cells causes depletion of CycT1. As presented in *Figure 6F*, CycT1 expression in activated primary T cells from a donor began to decrease at 24 hr (panel 1, compare lane 2 to lane 1, ~1.8-fold reduction) after the addition of ionomycin (1 μM) and continued at 48 hr (panel 1, compare lane 3 to lane 1, ~4.5-fold reduction) and 72 hr (panel 1, compare lane 4 to lane 1, ~9.1-fold reduction). Levels of CDK9 were largely unaffected (*Figure 6F*, panel 2, lanes 1–3). Similar changes were also observed with donor 2 treated with

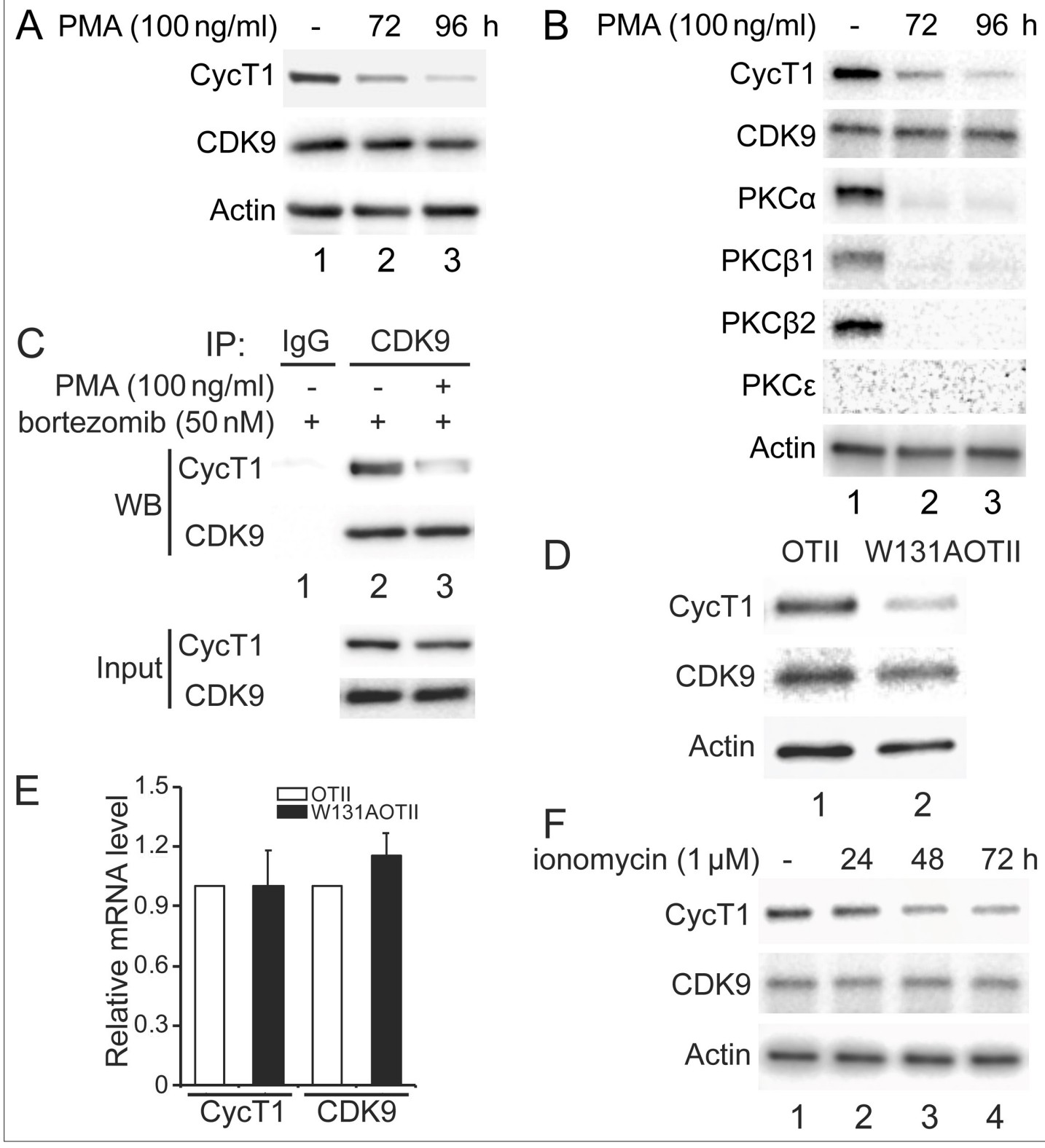

**Figure 6.** Depletion of PKCs leads to decreased levels of CycT1 in cell lines and primary cells. (**A**) Prolonged PMA treatment decreases levels of CycT1 in Jurkat cells. Jurkat cells were untreated (lane 1) or treated with 100 ng/ml PMA for 72 and 96 hr (lanes 2 and 3) before cell lysis. Panels 1 and 2 contain levels of endogenous CycT1 and CDK9 proteins, and panel 3 contains the loading control actin protein. (**B**) Prolonged PMA treatment decreases levels of CycT1 and PKC in activated primary CD4+ T cells. Activated primary CD4+ T cells were untreated (lane 1) or treated with PMA for 72 and 96 hr (lanes 2 and 3) before cell lysis. Panels 1–6 contain levels of endogenous CycT1, CDK9, PKCα, PKCβ1, PKCβ2, and PKCε proteins. Panel 7 contains the loading

*Figure 6 continued on next page*

*Figure 6 continued*

control actin protein. (**C**) Depletion of PKC impairs interactions between CycT1 and CDK9 in activated primary CD4+ T cells. Activated primary CD4+ T cells were treated with or without PMA (+/- signs on top) for 96 hr. At 72 hr, bortezomib was added for additional 24 hr before cell lysis. Co-IPs with CDK9 are presented in panels 1 and 2. Panels 3 and 4 contain input CycT1 and CDK9 proteins. (**D**) CycT1 levels are decreased in mouse anergic T cells. T cells were selected from WT OTII (WT ZAP70) or mutant W131AOTII (ZAP70W131A) mice and lysed. Panels 1 and 2 contain levels of endogenous CycT1 and CDK9 proteins. Lanes are: lane 1, WT OTII mice; lane 2, mutant W131AOTII mice. Panel 3 contains the loading control actin protein. (**E**) mRNA levels of CycT1 and CDK9 are equal in mouse WT and anergic T cells. Relative mRNA levels of CycT1 (left two bar graphs) and CDK9 (right two bar graphs) are presented as -fold change in W131AOTII T cells (black bars) above levels of WT OTII T cells (white bars). Error bars represent standard error of average (n = 3). (**F**) Prolonged ionomycin treatment decreases levels of CycT1 in activated primary CD4+ T cells. Activated primary CD4+ T cells were untreated (lane 1) or treated with 1 μM ionomycin for 24, 48, and 72 hr (lanes 2–4) before cell lysis. Panels 1 and 2 contain levels of endogenous CycT1 and CDK9 proteins. Panel 3 contains the loading control actin protein.

The online version of this article includes the following figure supplement(s) for figure 6:

**Figure supplement 1.** Chronic activation in primary cells decreases levels of endogenous CycT1 protein.

ionomycin (*Figure 6—figure supplement 1B*). Taken together, the absence of active PKC correlates with significant decreases of CycT1, which prevents the assembly of the functional P-TEFb complex.

## Discussion

In this study, we found that mutations of two critical residues in CycT1 (Thr143 and Thr149 to alanine) impair its binding to CDK9. Phosphorylation of these residues promotes CycT1:CDK9 interactions. Structural modeling indicated that phosphates on Thr143 and Thr149 in CycT1 increase intramolecular and intermolecular binding to specific residues in CycT1 and CDK9, respectively, which contributes to P-TEFb assembly. This prediction was confirmed experimentally. Thr143 and Thr149 are located in PKC consensus sequences. Indeed, PKC inhibitors inhibited CycT1 phosphorylation. Of PKC isoforms, PKCα and PKCβ were found not only to bind to CycT1 but to promote CycT1:CDK9 interactions and stabilize CycT1 in vivo and in vitro. Finally, depleting PKC or its inactivity led to CycT1 dissociation from CDK9 and its degradation in transformed cell lines as well as primary activated and anergic T cells. We conclude that the assembly and stability of P-TEFb are potentiated by the phosphorylation of CycT1, which is regulated by PKCα, PKCβ, and PP1.

Ours is the first study that addresses the reversible phosphorylation of CycT1 that helps to regulate P-TEFb. Based on previous mutageneses of CycT1, three threonine residues (Thr143, 149, and 155) could affect CycT1 stability and P-TEFb assembly. They are conserved in over 142 mammalian species and in CycT2 (sequences are picked out from NCBI and aligned). Moreover, previous reports indicated that these residues play critical roles in P-TEFb function (*Jadlowsky et al., 2008*; *Kuzmina et al., 2014*). Of these, we identified Thr143 and Thr149 to be key residues for P-TEFb assembly. Dephosphorylation of these residues was blocked by a high dose of okadaic acid, which inhibits PP1 and PP2A, or the general PP inhibitor calyculin A, but not by a low dose of okadaic acid, which inhibits only PP2A (*Figure 2A and B*). These results indicate that PP1, and not PP2A, is the candidate for CycT1 dephosphorylation. PP1 contains 3 catalytic and over 50 regulatory subunits, whose genetic inactivation is lethal to cells (*Cohen, 2002*; *Ferreira et al., 2019*). Indeed, they are phosphatases that regulate cell cycle cyclins, such as CycB, CycD1, and AIB1 (*Edelson and Brautigan, 2011*; *Ferrero et al., 2011*; *Vorlaufer and Peters, 1998*). As to kinases, PKCα and PKCβ isoforms phosphorylate these two threonine residues in CycT1 in vitro. Previous studies and our experiments including mutagenesis, using anti-phospho-threonine antibodies, in-gel Phospho-Tag staining, in vivo and in vitro PKC binding assays, and PKC inhibition strongly indicate that both Thr143 and Thr149 of CycT1 are involved in P-TEFb assembly. Further studies including mass spectrometry and/or specific anti-CycT1 phospho-peptide antibodies could confirm our findings. However, given that these two residues are adjacent to each other, it would complicate these further studies. Nonetheless, since its degradation and assembly have to occur in most if not all cells of the organism, the involvement of ubiquitously expressed and redundant kinases and phosphatases in this post-translational regulation of P-TEFb is not unexpected. Importantly, our study also reveals that the depletion of PKC, as occurs after chronic activation or phorbol ester treatment, results in P-TEFb disassembly and CycT1 degradation, which explains cells becoming unresponsive to external stimuli. It also explains why Tat, whose coactivator is P-TEFb, no longer functions in such cells (*Jakobovits et al., 1990*). Anergic cells also lack P-TEFb

(*Figure 6D*), which contributes to their unresponsive phenotype. From our and other studies, we also know that resting and memory T cells lack P-TEFb (*Garriga et al., 1998*; *Ghose et al., 2001*). Thus, proviral latency is maintained in these cells (*Fujinaga and Cary, 2020*). This situation most likely pertains to other DNA viruses, such as HSV and HTLV1/2, in yet other resting or terminally differentiated cells (*Kulkarni and Bangham, 2018*; *Nicoll et al., 2012*).

Although unphosphorylated CycT1 dissociates from P-TEFb and is degraded, CDK9 remains and is stabilized by HSP70 and HSP90 (*O'Keeffe et al., 2000*). This observation presents uncanny similarities to the regulation of cell cycle cyclins/CDKs, whose levels are also regulated by reversible phosphorylation, disassembly, and degradation. For example, CycE unbound to its kinase partner CDK2 is rapidly degraded via SCF E3 ligase-mediated proteasomal pathway (*Clurman et al., 1996*). Thus, the transcriptional cyclins/CDKs display and mimic regulatory paradigms of cell cycle cyclins/CDKs. This regulation is also very different from the P-TEFb equilibrium in growing, proliferating cells, where 7SK snRNP plays the major role. There, P-TEFb partitions between the active free state, bound to activators and/or the super-elongation complex, and the inactive state, where 7SK snRNA coordinates its sequestration with HEXIM1/2, LaRP7, MePCE in the 7SK snRNP. In this RNP, HEXIM1 or HEXIM2 inhibit CDK9 by binding to its ATP pocket, a situation that is reminiscent of CDK2 inhibition by p21 (*Russo et al., 1996*). Thus, our study has revealed an additional important aspect of the regulation of transcriptional elongation in cells.

Global mRNA sequencing studies examine changes in levels of specific transcripts between and in different states, that is, activation, proliferation, and differentiation of cells (*Sung and Rice, 2009*). However, levels of CycT1 and CDK9 mRNAs do not vary between resting and activated T cells (*Marshall et al., 2005*), between responsive and anergic T cells or exhausted T cells. This finding is important as these global studies do not look at levels of resulting proteins. Thus, they underestimate contributions of P-TEFb and/or other transcriptional CDKs in their evaluations of target genes. Additionally, P-TEFb must be recruited by transcription factors to the paused RNAPII at promoter-proximal regions. Indeed, P-TEFb can mediate short and long distance interactions between enhancers and promoters to promote transcription elongation and co-transcriptional processing of individual and clusters of genes (*Taube et al., 2002*). In this scenario, the C-terminal His-rich region of CycT1 interacts directly with the CTD of RNAPII (*Taube et al., 2002*). Thus, these post-translational modifications of P-TEFb play critical roles in activated transcription and for growth and proliferation of cells. In quiescent cells, only basal transcription is detected (*Cheung and Rando, 2013*; *Roche et al., 2017*). Importantly, external stimuli can reverse this phenotype via this reversible phosphorylation of P-TEFb. As a result, effects of potent activators, such as cMyc, NF-κB, steroid hormones, CIITA, etc., are translated to the productive transcription of their target genes (*Fujinaga, 2020*).

Finally, since our study revealed families of kinases and phosphatases that affect P-TEFb, it is possible that the use of more targeted phosphatase inhibitors could block this transition to quiescence and terminal differentiation of cells. Their use might even prevent the establishment of proviral latency. Although the phosphomimetic substitution of one of these two threonine residues was not successful (data not presented), it is possible that further modifications of CycT1 and CDK9 could create a constitutively active P-TEFb complex. If successful, modified P-TEFb complexes could be studied for effects on immune responses as well as latency induction and reversal in many different scenarios. The other approach would be to identify the E3 ligase that is responsible for the degradation of the unphosphorylated CycT1 protein. To this end, bortezomib and other proteasomal inhibitors have been examined already for the reversal of HIV latency in resting CD4+ T cells (*Cary and Peterlin, 2020*; *Li et al., 2019*). Taking all these findings into account, the understanding of complex post-translational regulation of P-TEFb promises to reveal additional approaches not only to proviral latency, but to host immune responses, cellular regeneration and dedifferentiation.

## Materials and methods

**Key resources table**

| Reagent type (species) or resource | Designation | Source or reference | Identifiers | Additional information |
|---|---|---|---|---|
| Antibody | CycT1 (E-3) (mouse monoclonal) | Santa Cruz Biotechnology | sc-271348 | WB (1:200) |

*Continued on next page*

*Continued*

| Reagent type (species) or resource | Designation | Source or reference | Identifiers | Additional information |
|---|---|---|---|---|
| Antibody | CDK9 (F-6) (mouse monoclonal) | Santa Cruz Biotechnology | sc-376646 | WB (1:200); IP 4 µg per test |
| Antibody | PKCα (H-7) (mouse monoclonal) | Santa Cruz Biotechnology | sc-8393 | WB (1:200) |
| Antibody | PKCβ1 (E-3) (mouse monoclonal) | Santa Cruz Biotechnology | sc-8049 | WB (1:200) |
| Antibody | PKCβ2 (F-7) (mouse monoclonal) | Santa Cruz Biotechnology | sc-13149 | WB (1:200) |
| Antibody | PKCε (E-5) (mouse monoclonal) | Santa Cruz Biotechnology | sc-1681 | WB (1:200) |
| Antibody | CDK9 (EPR22956-37) (rabbit polyclonal) | Abcam | ab239364 | WB (1:2000); IP 4 µg per test |
| Antibody | HA (clone HA-7) (mouse monoclonal) | Sigma-Aldrich | H3663 | WB (1:1000); IP 4 µg per test |
| Antibody | Flag (clone M2) (mouse monoclonal) | Sigma-Aldrich | F1804 | WB (1:1000); IP 4 µg per test |
| Antibody | HA (rabbit polyclonal) | Sigma-Aldrich | H6908 | WB (1:1000); IP 4 µg per test |
| Antibody | Flag (rabbit polyclonal) | Sigma-Aldrich | F7425 | WB (1:1000); IP 4 µg per test |
| Antibody | CycT1 (D1B6G) (rabbit monoclonal) | Cell Signaling Technology | 81464S | WB (1:1000) |
| Antibody | β-Actin (13E5) (rabbit monoclonal) | Cell Signaling Technology | 4970S | WB (1:5000) |
| Antibody | Phospho-threonine (42H4) (mouse monoclonal) | Cell Signaling Technology | 9386S | WB (1:500) |
| Antibody | Phosphoserine (rabbit polyclonal) | Abcam | ab9332 | WB (1:500) |
| Antibody | Phospho-tyrosine (P-Tyr-100) (mouse monoclonal) | Cell Signaling Technology | 9411S | WB (1:500) |
| Antibody | Normal rabbit control IgG | Santa Cruz Biotechnology | sc-2027 | IP 4 µg per test |
| Antibody | Normal mouse control IgG | Santa Cruz Biotechnology | sc-2050 | IP 4 µg per test |
| Antibody | Amersham ECL Mouse IgG, HRP-linked whole Ab (from sheep) | Cytiva | NA931 | WB (1:10,000) |
| Antibody | Amersham ECL Rabbit IgG, HRP-linked whole Ab (from donkey) | Cytiva | NA934 | WB (1:10,000) |
| Chemical compound, drug | Bortezomib | Calbiochem | 179324-69-7 | |
| Chemical compound, drug | Okadaic acid | Cell Signaling Technology | 5934S | |
| Chemical compound, drug | Staurosporine | Selleckchem | S1421 | |
| Chemical compound, drug | Sotrastaurin | Selleckchem | S2791 | |
| Chemical compound, drug | Bisindolylmaleimide IX | Selleckchem | S7207 | |

*Continued on next page*

*Continued*

| Reagent type (species) or resource | Designation | Source or reference | Identifiers | Additional information |
|---|---|---|---|---|
| Chemical compound, drug | HBDDE | Selleckchem | ab141573 | |
| Chemical compound, drug | H-7 | Abcam | ab142308 | |
| Chemical compound, drug | Cycloheximide (CHX) | Sigma-Aldrich | C4859 | |
| Chemical compound, drug | Bisindolylmaleimide IV | Selleckchem | S0754 | |
| Chemical compound, drug | Phorbol 12-myristate 13-acetate (PMA) | Sigma-Aldrich | P8139 | |
| Chemical compound, drug | Ionomycin | Sigma-Aldrich | I9657 | |
| Chemical compound, drug | MEK 1/2 inhibitor (MEK 1/2i) | Calbiochem | 444967 | |
| Chemical compound, drug | Enzastaurin | Selleckchem | S1055 | |
| Chemical compound, drug | VTX-27 | Selleckchem | S0069 | |
| Chemical compound, drug | Calyculin A | Cell Signaling Technology | 9902S | |
| Peptide, recombinant protein | BSA | Invitrogen | AM2616 | |
| Peptide, recombinant protein | β-Casein | Sigma-Aldrich | C6905 | |
| Peptide, recombinant protein | Recombinant human PKCα protein (active) | Abcam | ab55672 | |
| Peptide, recombinant protein | Recombinant human PKCβ1 protein (active) | Abcam | ab60840 | |

## Plasmids, antibodies, chemicals, and proteins

HA-CycT1 (h:CycT1), CDK9-Flag (CDK9:f), and plasmids containing mutated CycT1 or CDK9 sequences were constructed by cloning PCR fragments containing the coding sequences of CycT1 and CDK9 into pcDNA3.1 vector with indicated epitope tags. PKC plasmids (PKCα, β, γ, δ, ε, and θ) were obtained from Addgene, and their coding sequences were subcloned into the pcDNA3.1 vector containing the Flag epitope tag.

All antibodies, chemicals, and proteins are listed in the Key resources table.

## Cell culture

Human embryonic kidney (HEK) 293T cells were cultured in Dulbecco's modified Eagle's medium (DMEM) (Corning) with 10% fetal bovine serum (FBS) (Sigma-Aldrich), Jurkat cells and peripheral blood mononuclear cells (PBMCs) were cultured in Roswell Park Memorial Institute (RPMI) 1640 (Corning) with 10% FBS at 37°C and 5% $CO_2$. Resting CD4+ T cells were purified from bulk PBMCs by using Dynabeads Untouched Human CD4 T Cells Kit (Thermo Fisher Scientific). Selected CD4+ T cells were activated by Dynabeads Human T-Activator CD3/CD28 kit (Thermo Fisher Scientific) and were maintained in RPMI 1640 with 10% FBS, containing 30 U/ml IL-2 (Sigma-Aldrich).

## Cell manipulation

Transfection of plasmid DNA was conducted in 293T cells using Lipofectamine 3000 (Life Technologies) and X-tremeGENE HP DNA Transfection Reagent (Roche) according to the manufacturer's instructions.

293T cells were treated with 2 µM bortezomib for 12 hr, Jurkat cells were treated with 100 nM bortezomib for 12 hr, and activated CD4+ T cells were treated with 50 nM bortezomib for 12 hr before the cell lysis. 293T cells were treated with 5 nM or 1 µM okadaic acid for 1.5 hr, or with 150 nM calyculin A before the cell lysis. 293T cells, Jurkat cells, and activated CD4+ T cells were treated with MEK 1/2 inhibitor (MEK 1/2i) or different PKC inhibitors for 12 h before the cell lysis.

## Co-IP and quantification of WBs

293T, Jurkat, or CD4+ T cells were lysed on ice using RIPA buffer (50 mM Tris-HCl, pH 8.0, 5 mM EDTA, 0.1% SDS, 1.0% Nonidet P-40, 0.5% sodium deoxycholate, 150 mM NaCl) supplemented with the protease and phosphatase inhibitors, then for one-time sonication (level 4, 2 s), followed by a 10 min centrifugation (21,000 × $g$). The supernatant was precleared and incubated with indicated primary antibodies or control IgG overnight. Mixtures were then incubated with protein G-Sepharose beads for additional 2 hr, followed by five times' wash with RIPA buffer (500 mM NaCl). Co-IP samples and input (1% of whole-cell lysates) were subjected to WB as described previously (*Huang et al., 2018*).

Phosphorylation of CycT1 was detected by co-IP or IP, followed by WB using anti-pThr/Ser/Tyr antibodies. Then the membranes were stripped and reblotted with anti-HA antibodies to detect expression of h:CycT1 (unphosphorylated and phosphorylated forms). The band shift of the phosphorylated CycT1 proteins was confirmed by the actual size change in comparison to untreated conditions (no upper/phosphorylated bands detected).

WBs were visualized by enhanced chemiluminescence (ECL) (PerkinElmer) produced by HRP-conjugated secondary antibodies, and chemiluminescent signals were directly captured by LI-COR image analyzer. Band intensities of WBs were quantified using Image Studio software (LI-COR). Relative protein expression in whole-cell lysates was calculated by normalizing the indicated proteins with loading control β-actin. Relative protein-protein interactions in IP and co-IP were calculated by normalizing the IPed proteins with indicated antibodies targeted proteins. Quantification data were presented as fold change over values obtained with control samples.

## Trypsin digestion, in-gel silver staining, and phospho-staining

293T cells were lysed with RIPA buffer on ice. Lysates were sonicated once (level 4, 2 s) and cleared by centrifugation (21,000 × $g$, 10 min). The supernatant was precleared and incubated with HA-Ab-conjugated beads for 4 hr, followed by washing with RIPA buffer (500 mM NaCl) for three times and twice with RIPA buffer without detergents. Proteins associated with anti-HA-beads were eluted by incubation with the HA peptide (1 mg/ml). The eluted proteins were subjected to trypsin digestion using Rapid Trypsin kit (Promega) according to the manufacturer's instructions and separated in SDS-PAGE gel for silver staining by using Pierce Silver Stain Kit (Thermo) or phospho-staining by using Phospho-Tag Phosphoprotein Gel Stain Kit (ABP Biosciences) according to the manufacturer's instructions.

## In vitro PKC kinase assay

WT CycT1(280), mutant CycT1(280)TT143149AA, and EGFP proteins were expressed in 293T cells, isolated by HA-Ab-conjugated beads, and eluted from the beads by the HA-peptide (1 mg/ml). Eluted proteins were incubated with or without purified constitutively active PKC proteins (4 µg PKCα and PKCβ1) at 30°C for 1 hr in kinase buffer (20 mM HEPES, pH 7.4, 1.5 mM CaCl$_2$, 1 mM dithiothreitol, 10 mM MgCl$_2$, 5 mM ATP). Reaction mixtures were then subjected to SDS-PAGE gel, followed by in-gel Phospho-Tag staining.

## W131AOTII and control OTII cells preparation

W131AOTII mice were described previously (*Hsu et al., 2017*). Control OTII TCR transgenic mice were purchased from The Jackson Laboratory. Peripheral naïve (CD44$^{low}$CD62L$^+$) CD25$^-$Va2$^+$CD4$^+$ T cells were sorted from combined lymphoid organs (spleens and lymph nodes) of OTII or W131AOTII mice (8–12 weeks of age). The cells (10$^6$ cells) were washed with PBS, the supernatant was aspirated, and the pelleted cells were lysed as described above, then subjecting to WB assay.

## MD simulations and MM-GBSA binding energy calculations

To understand the role of phosphorylation of CycT1 residues Thr143 and Thr149, we performed all-atoms MD simulations of four CycT1:CDK9 complexes: WT, CycT1 phosphorylated at Thr143 and Thr149 (PThr$_{143,149}$), CycT1 phosphorylated at Thr143 (PThr$_{143}$), and CycT1 phosphorylated at Thr149 (PThr$_{149}$). Models of the four P-TEFb complexes were built based on the crystallographic structure of the human P-TEFb complex (PDB code 3MI9) using the AmberTools leap tool. All systems were neutralized with Na$^+$ and Cl$^-$ ions and solvate is a cubic box with periodic boundary conditions. Simulations were performed using the AMBER19 force-field (*Case et al., 2005*) and TIP3P water model (*Jorgensen et al., 1983*).

Simulations of each system were conducted according to the following protocol. First, each system was energy minimized for 5000 steps using the steepest-descent method while constraining solute non-hydrogen atoms with a 10 kcal/mol/Å$^2$ harmonic potential, followed by 5000 steps without restraints. Second, systems were heated from 10 K to 303.15 K for 50 ps in the NVT ensemble with heavy atom restraints applied. The temperature was controlled by Langevin dynamics, with a collision frequency of 5 ps$^{-1}$ (*Pastor et al., 2006*). Third, positional restraints were slowly released and densities were equilibrated over the course of 500 ps in the NPT ensemble at a temperature of 303.15 K and pressure conditions of 1 atm. The pressure was controlled using a MC barostat. Finally, unrestrained production simulations in the NPT ensemble were run for 500 ns for each system, yielding a cumulative sum of 2 µs of simulation time across systems. All calculations were carried out using an integration step of 2 fs. The SHAKE algorithm was applied to all hydrogen-containing bonds. MD simulations were conducted using the pmemd engine, with CUDA acceleration (*Salomon-Ferrer et al., 2013*).

For each production trajectory, the AmberTools *cpptraj* module was used to calculate root-mean-square deviations (RMSD) and root mean-squared fluctuation (RMSF) to monitor system equilibration and assess the local flexibility of the systems, respectively.

The binding energies of the four P-TEFb complexes were calculated by the MM-GBSA method (*Genheden and Ryde, 2015*; *Hou et al., 2011*; *Massova and Kollman, 2000*; *Miller et al., 2012*), implemented in Ambertools (*Case et al., 2005*). The MM-GBSA method has been used in many studies to calculate the binding energy of ligands with biomolecules as well as between biomolecules (*Hengel et al., 2016*; Malinvernii, *Malinverni et al., 2017*; *Zuo et al., 2019*). For each system, the MM-GBSA calculations were performed over an ensemble of 3000 snapshots extracted from the last 300 ns of the production MD trajectories using the program MMPBSA.py (*Miller et al., 2012*). We note that we did not include the entropic contributions in our calculations; however, the changes in binding energies can still be interpreted qualitatively (*Hou et al., 2011*).

## Acknowledgements

This study was supported by Damon Runyon Cancer Research Foundation Fellowship (to TTTN); NIH R01 AI049104 (FH, DCC, HP, BMP, and KF); NIH P01 AI091580 (to TTTN and AW); Howard Hughes Medical Institute (to TTTN and AW); Nora Eccles Treadwell Foundation (FH, HP, and KF); HARC center (NIH P50AI150476) (to FH, DCC, HP, BMP, KF, RR, IE, and AS). We thank Zeping Luo (lab member) for excellent technical assistance.

## Additional information

### Funding

| Funder | Grant reference number | Author |
|---|---|---|
| National Institute of Allergy and Infectious Diseases | R01 AI049104 | Fang Huang<br>Daniele C Cary<br>Hana Paculova<br>Boris Matija Peterlin<br>Koh Fujinaga |
| National Institute of Allergy and Infectious Diseases | P01 AI091580 | Trang TT Nguyen<br>Arthur Weiss |

| Funder | Grant reference number | Author |
|---|---|---|
| National Institute of Allergy and Infectious Diseases | P50AI150476 | Fang Huang<br>Ignacia Echeverria<br>Hana Paculova<br>Andrej Sali<br>Boris Matija Peterlin<br>Koh Fujinaga<br>Daniele C Cary<br>Ramachandran Rakesh |
| Howard Hughes Medical Institute | | Trang TT Nguyen<br>Arthur Weiss |
| Damon Runyon Cancer Research Foundation | | Trang TT Nguyen |
| Nora Eccles Treadwell Foundation | | Fang Huang<br>Koh Fujinaga |

The funders had no role in study design, data collection and interpretation, or the decision to submit the work for publication.

## Author contributions

Fang Huang, Methodology, Data curation, Conceptualization, Investigation, Methodology, Investigation, Supervision, Validation, Visualization, Writing – review and editing; Trang TT Nguyen, Data curation, Funding acquisition, Investigation, Methodology; Ignacia Echeverria, Data curation, Funding acquisition, Investigation, Methodology, Writing – review and editing; Ramachandran Rakesh, Data curation, Investigation, Methodology, Writing – review and editing; Daniele C Cary, Hana Paculova, Methodology, Writing – review and editing; Andrej Sali, Arthur Weiss, Funding acquisition, Investigation, Methodology, Writing – review and editing; Boris Matija Peterlin, Koh Fujinaga, Methodology, Data curation, Conceptualization, Funding acquisition, Investigation, Methodology, Project administration, Supervision, Visualization, Writing – review and editing

## Author ORCIDs

Fang Huang http://orcid.org/0000-0002-0511-2568
Ignacia Echeverria http://orcid.org/0000-0003-4717-1467
Andrej Sali http://orcid.org/0000-0003-0435-6197
Arthur Weiss http://orcid.org/0000-0002-2414-9024
Boris Matija Peterlin http://orcid.org/0000-0003-3692-219X
Koh Fujinaga http://orcid.org/0000-0003-4242-9078

## Decision letter and Author response

Decision letter https://doi.org/10.7554/eLife.68473.sa1
Author response https://doi.org/10.7554/eLife.68473.sa2

# Additional files

## Supplementary files

- Transparent reporting form
- Source data 1. Source data for figures and figure supplements.

## Data availability

Original data has been provided as Source data 1.

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
