## [Editor Report]

This study addresses an important question regarding the regulation of positive transcription elongation factor b (P-TEFb), which is an important regulator of gene expression targeted by the HIV Tat protein. The authors propose a novel mechanism with the potential to explain the differences in P-TEFb regulation between proliferating and quiescent cells, which might in turn have important implications for antiviral therapy.

---

## [Decision Letter]

**Decision letter after peer review:**

Thank you for submitting your article "Assembly and levels of P-TEFb depend on reversible phosphorylation of cyclinT1" for consideration by *eLife*. Your article has been reviewed by 3 peer reviewers, including Eric J Wagner as Reviewing Editor and Reviewer #1, and the evaluation has been overseen by Kevin Struhl as the Senior Editor.

Essential revisions:

1) Test the effect of phosphomimetic mutants because if these mutants function as predicted, it could add significant strength to the study.

2) provide stronger evidence that Thr143 and Thr149 are indeed phosphorylated in vitro and in vivo.

*Reviewer #1:*

In proliferating cells, the levels of p-TEFb comprised of CDK9 and Cyclin T1 are high to coincide with ongoing transcriptional activation. However, in resting cells the levels of Cyclin T1 are vanishingly low but the mechanism underlying this is poorly understood. The authors set out to understand how Cyclin T1 stability is regulated and uncovered a key role for two specific threonine, which are subject to phosphorylation by PKC and de-phosphorylation by PP1. The major strengths of the paper include determining that phosphorylation of the two threonines in Cyclin T1 not only contribute to its stability but are required for interaction with CDK9. Moreover, the authors show that PKC activity is a key regulated component in the transition from resting to activation that triggers p-TEFb assembly. The implication of this work is that it provides an explanation for long-sought question as to how transcriptional activation potential is regulated in specific cell types – notably memory T cells.

Overall, the data presented is of high quality and the presentation of the results is logical and well-described. In my view, the conclusions drawn are justified by the data presented. My suggestions below are meant to only strengthen the manuscript further.

Figure 2 – referring to 'panels' in the text is confusing as it was unclear what panel 1/2/3 were specifically referring to

Figure 2- the authors state that the pThr signal overlayed onto the upper band for CyCT1 but this is not easily ascertained based upon how the figures are constructed

Figure 2- the statement that OA enhanced co-IP by ~5 fold for the full length and truncation mutant is not convincing as stated. First of all, it doesn't look like OA enhanced CyCT coIP much at all in panel 2A and in 2B, that enhancement appears minor. Can the authors state explicitly how this was quantified and if it was done so using replicate IPs so that reproducibility can be determined?

Figure 2C – The T143/T149A mutant is convincing that the pThr signal is diminished. But it is confusing why the input CycT1 signal in the T143/T149AA is identical to the wt-280 protein ('input' CyCT1 bottom blot, lanes 3vs4). This would suggest that mutation of these two threonines does not substantively alter overall phosphorylation of the 280 cycT1 protein. This result almost seems ignored, can the authors clarify this interpretation?

Figure 3A: the authors could consider a slightly different or additional perspective in their image. It is difficult to see the K68 interaction with T149 as the side chains are right on top of each other.

Figures3B/C/D: similar comment to above in terms of how the fold reduction in binding is calculated. The data in panel C/D look very convincing but the data in panel B is not as robust – yet the 'fold-reduction' is calculated as very close for B/C?

Figure 4: overall, the data shown is quite compelling but there is one overarching concern I have. The data in 4C demonstrates that treatment with staurosporine eliminates any pThr signal present on CycT1. While this is convincing, it is also apparent that all aberrantly migrating CycT1 that appears after OA treatment is collapsed into a single band suggesting that all phosphorylation of CycT1 is sensitive to staurosporine. This doesn't seem likely given the authors showed in panel 3C (bottom) that TT143,149AA, which diminishes pThr signal has no apparent impact to the remaining phosphorylations of CycT1 as the aberrant migrating form is unchanged. This speaks to the concern of staurosporine specificity but could be mitigated if the authors also conduct a PKC RNAi experiment followed by transfection of their HA-CycT1.

*Reviewer #2:*

This study addresses an important question regarding the regulation of positive transcription elongation factor b-P-TEFb, the complex of cyclin-dependent kinase 9 (CDK9) and cyclin T1. In proliferating cells, this complex is the major rate-limiting regulator of transcription elongation by RNA polymerase II (RNAPII) and target of HIV Tat protein-which co-opts P-TEFb to drive expression of viral genes-and is negatively regulated by an inhibitory 7SK ribonucleoprotein (RNP) complex. This axis does not appear to operate in quiescent cells, including resting or exhausted T cells that are refractory to Tat, apparently because of decreased levels of cyclin T1 protein despite high levels of the mRNA. The authors propose a novel regulatory mechanism with the potential to explain the differences in P-TEFb regulation between proliferating and quiescent cells: phosphorylations on two Thr residues in cyclin T1 that stabilize its binding to Cdk9, which are placed by protein kinase C (PKC) isoforms and removed by protein phosphatase 1 (PP1), and which become dephosphorylated in quiescent cells, leading to disassembly of the P-TEFb complex and subsequent proteasomal degradation of the free cyclin. This is an attractive hypothesis, both because it would solve a longstanding puzzle in the field and yield insights into potential antiviral drug targets, and because the proposed mechanism of CDK regulation it posits is novel and surprising (possibly unprecedented). The data are for the most part consistent with the proposed model but fall short of proving it; many of the functions of the individual players or modifications, and relationships between them, are inferred rather than demonstrated. There are questions of specificity-about the kinase and phosphatase inhibitors used, and about the phosphorylated residues being targeted-that would need to be answered more definitively before this mechanism can be considered validated. Below I list my major, specific concerns and comments:

1. The implication of PKC and PP1, the two key enzymatic players in the P-TEFb regulatory network proposed here, rests heavily on experiments with inhibitory small molecules that are at best only partially selective for their intended targets. For example, staurosporine, the "PKC inhibitor" used for most of their experiments, is a relatively promiscuous protein kinase inhibitor. Likewise, okadaic acid is a potent inhibitor of PP2A with reduced but significant potency towards PP1 and other phosphoprotein phosphatase (PPP) family members; the authors' argument for its PP1-specificity in this setting is that it influences cyclin T1 stability at high but not low concentrations. I do not find this reasoning compelling; at the higher concentrations needed to inhibit PP1 in vivo, PP2A would also be more completely inactivated.

2. The use of dominant negative, kinase-dead variants of PKC somewhat allays the concerns regarding specificity of the kinase inhibitors, and suggests that PKC activity might indeed be influencing cyclin T1 phosphorylation state, but a demonstration that PKC can directly phosphorylate cyclin T1, specifically on Thr143 and Thr149, is missing, and would be needed to justify some of the stronger conclusions drawn here (e.g. on p. 10, p. 14 and p. 15, last sentence of first paragraph in each case).

3. There is in fact no demonstration that Thr143 and Thr149 are actually phosphorylated in vivo or in vitro, only that their mutation to Ala diminishes reactivity of cyclin T1 with anti-pThr antibodies. This could be an indirect effect on phosphorylation of a different residue due to conformational changes caused by the mutations. The authors state, on p. 20 of Discussion, that these sites "were missed" in previous studies but to justify that assertion would need to show unambiguously that they are indeed phosphorylated (i.e., not merely the computational prediction in Figure S1 and the circumstantial evidence obtained from mutagenesis studies). Importantly, recombinant, highly active and apparently stable P-TEFb can be purified-and crystallized-in the absence of these phosphorylations (e.g. in Baumli et al., 2008). There is precedent for phosphorylations (on the CDK) being required to stabilize CDK-cyclin complexes in vivo but dispensable in vitro or in overexpression conditions, but to conclude that is the case here would require proof that these residues are indeed phosphorylated in vivo.

4. There is a circularity to the logic behind the choice of PKC isoforms to target specifically (p. 13): Thr143 and Thr149 looked like PKC sites, leading the authors to test a panel of PKC inhibitors that were specific for the isoforms they selected for further studies. A more cogent argument would be based on both positive and negative data, i.e., inhibitors of different isoforms (and different kinases entirely) not having the same effects.

5. Selectivity in phosphatase inhibition is notoriously hard to achieve. Nonetheless, there are small molecules with some selectivity for PP1 over PP2A (e.g. tautomycetin), greater selectivity for PP2A (e.g. cytostatin) or ~equal potency towards both (e.g. calyculin A); the authors should try these inhibitors and/or targeted depletion of PP1 catalytic (or regulatory) subunits to test PP1 involvement more rigorously.

6. The authors should test potentially phosphomimetic substitutions of these residues (e.g. T143E, T149E)-a manipulation they allude to in the Discussion but did not attempt. Although Glu is not guaranteed to mimic pThr, the upside if it does would be considerable: if a constitutive negative charge is sufficient to stabilize P-TEFb, it should increase cyclin T1 in quiescent cells (perhaps conferring sensitivity to Tat) and, if the proposed model is correct, make cyclin T1 in proliferating cells resistant to degradation induced by PKC inhibition.

7. The study would be greatly strengthened by measurements of CDK9 activity after the various manipulations tested, i.e., cyclin T1 mutations, PKC ablation, PP1 inhibition.

8. The effectiveness of PKC inhibitors or dominant-negative alleles should be gauged with measurements of phosphorylation on known PKC substrates, if phosphospecific antibodies are available for these targets.

9. The second sentence of the abstract is ambiguously worded: it should be made clearer that what is absent in quiescent cells is P-TEFb, not 7SK RNP (if that is what the authors mean).

10. Pp. 2-3, non-expert readers might not appreciate distinction between promoter clearance and promoter-proximal pause release (and thus between functions of CDK7 and -9), which are conflated to a significant degree in the authors' description.

11. P. 3, first paragraph, sentence starting "CDK9 is…" "DRB" should be "DRB sensitivity inducing factor (DSIF)" (referred to in the next sentence).

12. P. 6 and elsewhere, the authors should avoid the nomenclature "T3A" and T2A" to refer to different combinatorial mutations (i.e., of two or three different Thr residues) because it will be confused with single point mutations, i.e., of "Thr3" or "Thr2" (which are nonexistent). I suggest the more conventional shorthand "3TA" and "2TA" to refer to these alleles.

13. I am just a bit skeptical about the quantification of immunoblot signals: some of the numbers reported for fold-differences do not match the visual evidence (i.e., the band intensities). They appear to be using ECL (rather than fluorescence) for detection and quantification, so these numbers should probably be taken with a grain of salt. I would recommend the authors provide more detail about the quantification, e.g. the raw data for the blot scanning, preferably with error bars.

14. There is a sentence fragment on p. 13, starting with "Since the catalytic domains…"

15. The precedent of cyclin E degradation cited on p. 20 is interesting but not quite apt: in that case degradation is promoted by phosphorylation of cyclin E (on a sequence motif known as a phosphodegron), and I am not aware of any requirement for cyclin E phosphorylation in binding to CDK2, so its relevance to the proposed mechanism of cyclin T1 degradation is not clear.

*Reviewer #3:*

The manuscript by Huang and colleagues describes two phosphorylations sites in human Cyclin T1, which regulate the interaction to CDK9 and contribute to the binding and stability of Cyclin T1 in quiescent and terminally differentiated cells. The author describe that the pool of free Cyclin T1 is rapidly degraded when it is not phosphorylated on these two respective sites in the cyclin boxes of cyclin T1. The PP1 phosphates inhibitor okadaic acid leads to P-TEFb stabilization, as less CycT1 is degraded, whereas kinase inhibitors targeting PKC result in P-TEFb disassembly and degradation of CycT1. Using a set of different kinase inhibitors, the authors find that PKC-α and PKC-β are responsible for CycT1 phosphorylation.

The findings described in this study bring the transcription regulating cyclins, as CycT1, closer to the cell cycle regulating cyclins, which are long known for the tight regulation between expression upregulation and ubiquitination downregulation. These findings are potentially important, as they could shift our view on the transcriptional cyclins and the need for transcriptional CDK regulators (as 7SK/Hexim1 and Brd4, HIV Tat, AFF4 …), if gene expression and degradation is indeed another layer of regulation of these cyclins.

There is one major conundrum to this study: Multiple biochemical and structural studies on CDK9 and Cyclin T1 have been performed, often by co-expression and co-purification of these two subunits. In none of them (to my knowledge) a phosphorylated T143 or T149 residue was identified. The sentence "Unphosphorylated CycT1 dissociates from P-TEFb and is degraded." (page 20 and similar in the abstract) seems therefore farfetched. If pT143 and pT149 stabilizes the complex in such a profound degree, why are researchers able to form the non-phosphorylated complex in the first place? CycT1 can be expressed from *E. coli* exhibiting no phosphorylation and binds properly well to CDK9. It would be very beneficial and in support of this study to know the dissociation constants for the interaction partners in a CycT1 phosphorylated and non-phosphorylated state.

I am surprised that a phosphorylated threonine mimetic, as glutamic acids (E) or aspartic acid (D), was not analyzed in these experiments. To this reviewers' opinion the authors should analyze how a double mutation of CycT1 (T143E/T149E) acts on CDK9 binding, P-TEFb complex formation and the stability of Cyclin T1. Using only the T143A/T149A mutant but not a phosphorylation mimetic is a lack in the design of the study.

Throughout the entire manuscript, the authors are unprecise and ambiguous with predictions (computational, structural, …) and 'real' experimental findings. E.g.: Page 18, Discussion, 3 sentence: "Structural analyses revealed that phosphates on Thr143 and Thr149 in CycT1 increase intramolecular and intermolecular binding to specific residues in CycT1 and CDK9, respectively, which potentiates P-TEFb assembly and stabilizes CycT1." In Figure 3A, a model is displayed but not an experimentally determined structure with phosphorylated residues, as 3MI9 does not contain any phosphorylation in Cyclin T1. The figure legend to 3A should clearly say this. The description in the main text is okay, but the text of the figure legend is clearly misleading and almost fraud. This is a model only!

Another example is on page 8: "An online database for phosphorylation site prediction (NetPhos 3.1, developed by Technical University of Denmark) scores Thr143 and Thr149 above the threshold value (default 0.5), indicating that these threonines are potential phosphorylation sites (Figure S1). To verify that Thr143 and Thr149 are the main phosphorylation sites in CycT1, …" The second sentence starting with "To verify …." suggests, that the first observation are real data but not simply predictions. Moreover, the prediction has been made on a 43-residue sequence but not the full CycT1 structure (Figure S1) as I understand. This does not account for any tertiary structure etc. and is therefore not to the highest standard. I would not consider this analysis good evidence for phosphorylation sites.

There are clearly dissociation constants missing for the statement that the threonine phosphorylation in cyclin T1 is increasing its binding capacity to CDK9 and the stability of the cyclin. From the model it seems that pT143 could also only act indirectly as it forms internal contacts within the cyclin but is not in the direct interface to the kinase.

[Editors’ note: further revisions were suggested prior to acceptance, as described below.]

Thank you for submitting your article "Assembly and levels of P-TEFb depend on reversible phosphorylation of cyclin T1" for consideration by *eLife*. Your article has been reviewed by 2 peer reviewers, and the evaluation has been overseen by a Reviewing Editor and Kevin Struhl as the Senior Editor. The following individual involved in review of your submission has agreed to reveal their identity: Matthias Geyer (Reviewer #3).

There was agreement among the Reviewers that the revised submission had been improved and the new data presented mostly addressed previously noted critiques. However, there was still a shared concern that stronger data is needed to demonstrate that these two threonine residues are phosphorylated in cells. In light of these counter-balancing thoughts, the Editors have decided on an additional round of revision to address this specific concern. We highlight essential revisions below and provide specific feedback from the Reviewers.

Essential revisions:

1) Conduct mass spectrometry on preferably full-length Cyclin T1 to demonstrate phosphorylation at T143 and T149 or provide convincing evidence for their phosphorylation using phospho-specific antibodies if such a reagent is in hand.

*Reviewer #2:*

This is a revision of a manuscript I reviewed previously. The authors have addressed one of my major concerns by providing evidence that PKC can phosphorylate a cyclin T1 fragment in T143/149-dependent fashion in vitro (Figure 5G). I am less persuaded by the evidence implicating PP1 in dephosphorylating cyclin T1, but this is largely owing to the technical difficulty of proving phosphatase specificity in cell-based experiments. Taken together, there is substantial reason to believe, based on the data presented, that presence or absence of PKC-dependent signaling is a determining factor in P-TEFb activity levels in different settings (e.g. resting or exhausted versus activated T cells).

My biggest concern remains, however, that the mechanism the authors propose for this regulation-phosphorylation of two specific residues on cyclin T1 that stabilize an interaction with CDK9-is only incrementally closer to being validated in the revised version. The new data addressing this point were obtained in a complicated setup, in which a severely truncated cyclin T1 fragment (missing residues needed for interaction with CDK9) was expressed in 293T cells and then pulled down and detected in gels by a commercial phosphoprotein-staining reagent, either intact (Figure 2D) or after tryptic digestion (2E). Mutation of T143 and T149 to A abolished the signal, which is promising but still preliminary evidence that these sites might be phosphorylated in intact cyclin T1 under physiologic conditions. This does not address the concern, also expressed by Reviewer 3, that this requirement-and the phosphorylations themselves-have escaped previous detection. The authors dismiss this concern by pointing out that structures of P-TEFb were solved for complexes expressed in insect cells, but this is not persuasive; in those structures, phosphorylation on T186 of the CDK9 activation segment was readily observed (Baumli et al., 2008; Tahirov et al., 2010). Fully active, stable P-TEFb complexes can also be reconstituted from purified CDK9 expressed in insect cells and cyclin T1 purified from bacteria. Finally, a quick check of the PhosphoSite online data base did not turn up either of these sites. So I must reiterate my initial position that, because the authors are proposing a novel and highly detailed mechanism for regulating an essential transcription regulator, the existence, necessity and sufficiency of T143 and T149 phosphorylation simply must be proven and not merely inferred. Typically, a study such as this would include dispositive supporting data obtained via mass spectrometry, phospho-specific antibodies, or both, to gain acceptance at a top-tier journal.

*Reviewer #3:*

The authors present a thoroughly revised version of their manuscript that addresses the main concerns from the previous review process. Particularly the second point regarding stronger evidence that T143 and T149 are directly phosphorylated is now supported by additional data. However, I miss any new data within the revised manuscript regarding the first main point on the use of phosphomimetic mutants in Cyclin T1. The authors state in their reply letter that the CycT1 T149D or T149E mutants maintained CDK9 binding, T143 to E or D instead did not. I think that this first mutant site T149(E/D) that sustains binding clearly strengthen their findings. Why don't they show these binding data, or did I miss it? The contribution of the T149 position to the interaction would be anyhow more indirect as shown from the modelling and it is well accepted, that these phosphomimetic mutants must not work. But as one of them seems to work, I am confused that these data are not contained in the manuscript. The new molecular dynamics simulations and MM-GBSA binding energy calculations are nice complementary data but cellular binding/precipitation data would be very convincing. The other points are well addressed.

---

## [Author Response]

Essential revisions:1) Test the effect of phosphomimetic mutants because if these mutants function as predicted, it could add significant strength to the study.

We created phosphomimetic mutants for Thr143 and Thr149. While Thr149 to aspartic or glutamic acid restored this binding, the same substitutions for Thr143 did not. Indeed, we agree with Reviewer 2 that not all phosphomimetic substitutions work, which also happened in our hands.

After investigating many published phosphorylation-related papers, there emerges no consistent picture for acidic amino acid substitutions to mimic phosphorylated serine, threonine or tyrosine residues. As we know, the charge densities, distributions and pKas of the carboxyl group of aspartic acid or glutamic acid are quite different from a phosphate group. There are some good examples such as MEK and PKB/Akt where such substitutions work quite well (to ~10-30% of fully phosphorylated wild type proteins). In other cases, it doesn't work in related proteins such as MKK4 (Thr234) (Khan M., et al., Brassinosteroid-regulated GSK3/Shaggy-like kinases phosphorylate mitogen-activated protein (MAP) kinase kinases, which control stomata development in *Arabidopsis thaliana*. J Biol Chem. 2013 Mar 15;288(11):7519-7527.). It’s also worth considering what phosphorylated amino acids are doing in a given protein. For example, hosphor-specific antibodies do not recognize substituted aspartates or glutamates – indicating quite different structural epitopes represented by these substitutions (perspectives given by James R Woodgett, Mount Sinai Hospital, Toronto). The most recent publication also demonstrated that phosphomimetic substitutions were not able to restore the activation loop of RSK (Somale, D., et al., Activation of RSK by phosphomimetic substitution in the activation loop is prevented by structural constraints. Sci Rep 10**,** 591 (2020)).

Consistent with our further Molecular Mechanics/Generalized Born Surface Area calculations (MM-GBSA) for the P-TEFb complex,, the predominant contribution to the increased binding energy of the complex comes from electrostatic (ΔE_elec_) and polar solvation (ΔE_solv−polar_) energies. This finding is consistent with the stabilizing interactions described in Figures 1 and 2. Thus, the phosphorylation of Thr143 and Thr149 in CycT1 is thermodynamically advantageous for interactions between CDK9 and CycT1 (Table S1).

2) Provide stronger evidence that Thr143 and Thr149 are indeed phosphorylated in vitro and in vivo.

In addition to existing IP-WBs and kinase inhibitors experiments that provideextensive confirmation of this phosphorylation, we performed additional in-gel phosphoprotein staining studies for the direct phosphorylation of Thr143 and Thr149. First, the WT CycT1(280) or CycT1(192) rather than their AA mutant counterparts contained significantly higher levels of phospho-threonines in cells (Figure 2D, Figure 2—figure supplement 1C). Second, in-gel phosphoprotein staining of a trypsin peptide from the WT CycT1(192) and its AA mutant counterpart demonstrated specific phosphorylation of Thr143 and Thr149 in cells (Figure 2E). Third, we present the direct PKC phosphorylation of Thr143 and Thr149 in vitro (Figure 5G). Taken together, existing and added data provide overwhelming evidence that Thr143 and Thr149 are phosphorylated in vivo and in vitro. They add substantially to all conclusions of our work.

We believe that the revised manuscript addresses all points raised by the reviewers and therefore is suitable for publication in *eLife*. Below are our point-by-point responses to each comment.

Reviewer #1:In proliferating cells, the levels of p-TEFb comprised of CDK9 and Cyclin T1 are high to coincide with ongoing transcriptional activation. However, in resting cells the levels of Cyclin T1 are vanishingly low but the mechanism underlying this is poorly understood. The authors set out to understand how Cyclin T1 stability is regulated and uncovered a key role for two specific threonine, which are subject to phosphorylation by PKC and de-phosphorylation by PP1. The major strengths of the paper include determining that phosphorylation of the two threonines in Cyclin T1 not only contribute to its stability but are required for interaction with CDK9. Moreover, the authors show that PKC activity is a key regulated component in the transition from resting to activation that triggers p-TEFb assembly. The implication of this work is that it provides an explanation for long-sought question as to how transcriptional activation potential is regulated in specific cell types – notably memory T cells.Overall, the data presented is of high quality and the presentation of the results is logical and well-described. In my view, the conclusions drawn are justified by the data presented. My suggestions below are meant to only strengthen the manuscript further.Figure 2 – referring to 'panels' in the text is confusing as it was unclear what panel 1/2/3 were specifically referring to.

We apologize for the confusion. In the revised manuscript, we added an explanation about panel numbers (top panels are panel 1, and the number increases from top to bottom).

Figure 2- the authors state that the pThr signal overlayed onto the upper band for CyCT1 but this is not easily ascertained based upon how the figures are constructed

We added more detailed explanation about how WBs were processed in the manuscript. To detect CycT1 pThr signals by IP-WB, we first stained with anti-pThr antibodies, then the membranes were stripped and reblotted with anti-HA antibodies to detect HA-CycT1 (unphosphorylated and phosphorylated forms). Also, the band shift of the phosphorylated CycT1 protein was confirmed by the actual size change in comparison to the untreated condition (no upper/phosphorylated bands detected). Combining all these data, the upper bands of CycT1 detected by anti-pThr antibodies represent phosphorylated CycT1 proteins (especially after the treatment with protein phosphatase inhibitors).

Figure 2- the statement that OA enhanced co-IP by ~5 fold for the full length and truncation mutant is not convincing as stated. First of all, it doesn't look like OA enhanced CyCT coIP much at all in panel 2A and in 2B, that enhancement appears minor. Can the authors state explicitly how this was quantified and if it was done so using replicate IPs so that reproducibility can be determined?

All WB images were captured by the LI-COR imaging system. Luminescent signals corresponding to each protein band were processed with their analysis program. They were within the linear range of measurements according to Li-COR instructions. Relative protein-protein interactions in co-IP were calculated by normalizing the IPed proteins with indicated antibodies targeted proteins. Quantification data were presented as fold changes over values obtained with control samples (values set as ‘1’) (more details were added in the ‘Materials and methods’). For example, in Figure 2A, relative CycT1:CDK9 interactions were calculated by normalizing intensities of IPed CycT1 bands with IPed CDK9 bands. Data are from three independent experiments with excellent reproducibility. In addition, we repeated experiments by using low and high concentrations of okadaic acid as well as calyculin A, which is another protein phosphatase inhibitor. These new co-IPs are provided in the revised Figures 2A and 2B.

Figure 2C – The T143/T149A mutant is convincing that the pThr signal is diminished. But it is confusing why the input CycT1 signal in the T143/T149AA is identical to the wt-280 protein ('input' CyCT1 bottom blot, lanes 3vs4). This would suggest that mutation of these two threonines does not substantively alter overall phosphorylation of the 280 cycT1 protein. This result almost seems ignored, can the authors clarify this interpretation?

Our interpretation of the data is that there are additional phosphorylation sites in CycT1 that are not involved in P-TEFb assembly and/or CycT1 degradation, and these phosphorylation sites are also stabilized by okadaic acid and/or calyculin A. They could contribute to the much clearer band shift of CycT1. Indeed, we compared levels of other phosphorylated residues between WT CycT1(280) and mutant CycT1(280)TT143149AA, and they were only slightly different between these two proteins, supporting this interpretation. New data in the revised manuscript address this issue more directly (see comments to editor, above).

Figure 3A: the authors could consider a slightly different or additional perspective in their image. It is difficult to see the K68 interaction with T149 as the side chains are right on top of each other.

We updated the structural analysis and figure in the revised manuscript (Figure 3A). They present a better view of the linkage between Thr149 in CycT1 and Lys68 in CDK9.

Figures3B/C/D: similar comment to above in terms of how the fold reduction in binding is calculated. The data in panel C/D look very convincing but the data in panel B is not as robust – yet the 'fold-reduction' is calculated as very close for B/C?

Band intensities were quantified as described above. Additionally, we also repeated experiments using the mutant CycT1 Q73A protein and the data are consistent with our conclusion. The WB in Figure 3B was replaced with new data in the revised manuscript.

Figure 4: overall, the data shown is quite compelling but there is one overarching concern I have. The data in 4C demonstrates that treatment with staurosporine eliminates any pThr signal present on CycT1. While this is convincing, it is also apparent that all aberrantly migrating CycT1 that appears after OA treatment is collapsed into a single band suggesting that all phosphorylation of CycT1 is sensitive to staurosporine. This doesn't seem likely given the authors showed in panel 3C (bottom) that TT143,149AA, which diminishes pThr signal has no apparent impact to the remaining phosphorylations of CycT1 as the aberrant migrating form is unchanged. This speaks to the concern of staurosporine specificity but could be mitigated if the authors also conduct a PKC RNAi experiment followed by transfection of their HA-CycT1.

Thanks for pointing out the concerns about the specificity of the PKC inhibitor staurosporine. First, staurosporine is a very strong and general PKC inhibitor, which can neutralize robustly increased phosphorylation effects by protein phosphatase inhibitors. Indeed, staurosporine removed all threonine phosphorylation and the resulting CycT1 proteins collapsed into one single band. As mentioned above, our interpretation is that there are multiple phosphorylation sites that are sensitive to staurosporine, and among them are Thr143 and Thr149 that are involved in P-TEFb assembly and CycT1 degradation. To obtain results with more specific PKC inhibitors, we employed two other PKC inhibitors (bisindomaleinide IX and HBDDE) and one non-PKC inhibitor MEK 1/2i which also inhibited interactions between CycT1 and CDK9, and diminished phosphorylation of Thr143 and Thr149 (Figure 4B and 4C), confirming that these residues are phosphorylated by PKC, which is important for P-TEFb assembly. Meanwhile, with other three PKC inhibitors (not inhibiting PKCα and PKCβ), only these selected PKC inhibitors were able to decrease the endogenous levels of CycT1 proteins in activated primary CD4^+^ T cells (Figure 4-figures supplement 1C, 1D and 1E). In addition, experiments with dominant negative PKCs and the new in vitro PKC kinase assay (Figure 5G) also support the involvement of PKC in this process.

Reviewer #2:This study addresses an important question regarding the regulation of positive transcription elongation factor b-P-TEFb, the complex of cyclin-dependent kinase 9 (CDK9) and cyclin T1. In proliferating cells, this complex is the major rate-limiting regulator of transcription elongation by RNA polymerase II (RNAPII) and target of HIV Tat protein-which co-opts P-TEFb to drive expression of viral genes-and is negatively regulated by an inhibitory 7SK ribonucleoprotein (RNP) complex. This axis does not appear to operate in quiescent cells, including resting or exhausted T cells that are refractory to Tat, apparently because of decreased levels of cyclin T1 protein despite high levels of the mRNA. The authors propose a novel regulatory mechanism with the potential to explain the differences in P-TEFb regulation between proliferating and quiescent cells: phosphorylations on two Thr residues in cyclin T1 that stabilize its binding to Cdk9, which are placed by protein kinase C (PKC) isoforms and removed by protein phosphatase 1 (PP1), and which become dephosphorylated in quiescent cells, leading to disassembly of the P-TEFb complex and subsequent proteasomal degradation of the free cyclin. This is an attractive hypothesis, both because it would solve a longstanding puzzle in the field and yield insights into potential antiviral drug targets, and because the proposed mechanism of CDK regulation it posits is novel and surprising (possibly unprecedented). The data are for the most part consistent with the proposed model but fall short of proving it; many of the functions of the individual players or modifications, and relationships between them, are inferred rather than demonstrated. There are questions of specificity-about the kinase and phosphatase inhibitors used, and about the phosphorylated residues being targeted-that would need to be answered more definitively before this mechanism can be considered validated. Below I list my major, specific concerns and comments:1. The implication of PKC and PP1, the two key enzymatic players in the P-TEFb regulatory network proposed here, rests heavily on experiments with inhibitory small molecules that are at best only partially selective for their intended targets. For example, staurosporine, the "PKC inhibitor" used for most of their experiments, is a relatively promiscuous protein kinase inhibitor. Likewise, okadaic acid is a potent inhibitor of PP2A with reduced but significant potency towards PP1 and other phosphoprotein phosphatase (PPP) family members; the authors' argument for its PP1-specificity in this setting is that it influences cyclin T1 stability at high but not low concentrations. I do not find this reasoning compelling; at the higher concentrations needed to inhibit PP1 in vivo, PP2A would also be more completely inactivated.

As mentioned above (to Reviewer 1), the revised manuscript, we added data with more specific PKC inhibitors including bisindomaleinide IX and HBDDE and a non-PKC kinase inhibitor (MEK1/2 inhibitor) as well as with another protein phosphatase inhibitor, calyculin A. They all support our conclusions. Moreover, we now include data with low concentrations of okadaic acid that have no effect on CycT1 phosphorylation and P-TEFb assembly. This concentration inhibits PP2A, not PP1. While we do agree with the reviewer's comment "at the higher concentrations needed to inhibit PP1 in vivo, PP2A would also be more completely inactivated", the fact that there is no such effect with low concentrations of okadaic acid clearly indicates that PP2A has a minor, if any, involvement in this process. Elucidation of PP isotypes involved in biological processes is very difficult because of the large numbers of isoforms and subunits and the lack of highly specific phosphatase inhibitors. However, using high vs low concentration of okadaic acid to separate between PP1 and PP2A is generally accepted by the field; indicated papers are listed below:

1). Pei JJ, et al., Okadaic-acid-induced inhibition of protein phosphatase 2A produces activation of mitogen-activated protein kinases ERK1/2, MEK1/2, and p70 S6, similar to that in Alzheimer’s disease. Am J Pathol. 2003.

2). Mailhes JB, et al,. Okadaic acid, an inhibitor of protein phosphatase 1 and 2A, induces premature separation of sister chromatids during meiosis I and aneuploidy in mouse oocytes in vitro. Chromosome Res. 2003.

3). Swingle M, Ni L, Honkanen RE. Small-molecule inhibitors of ser/thr protein phosphatases: specificity, use and common forms of abuse. Methods Mol Biol. 2007.

4). A. B. Dounay and C. J. Forsyth, “ Okadaic Acid: The Archetypal Serine / Threonine Protein Phosphatase Inhibitor”, Current Medicinal Chemistry 2002.

5). Takai A, et al., Inhibitory effect of okadaic acid derivatives on protein phosphatases. A study on structure-affinity relationship. Biochem J. 1992.

6). Lourdes Garcia, et al., PP1/PP2A phosphatases inhibitors okadaic acid and calyculin A block ERK5 activation by growth factors and oxidative stress. FEBS Letters. Volume 523, Issues 1–3, 2002.

7). Swingle M, Ni L, Honkanen RE. Small-molecule inhibitors of ser/thr protein phosphatases: specificity, use and common forms of abuse. Methods Mol Biol. 2007.

2. The use of dominant negative, kinase-dead variants of PKC somewhat allays the concerns regarding specificity of the kinase inhibitors, and suggests that PKC activity might indeed be influencing cyclin T1 phosphorylation state, but a demonstration that PKC can directly phosphorylate cyclin T1, specifically on Thr143 and Thr149, is missing, and would be needed to justify some of the stronger conclusions drawn here (e.g. on p. 10, p. 14 and p. 15, last sentence of first paragraph in each case).

We performed direct phosphorylation studies in vivo and in vitro (Figures 2 and 5). They are included in the revised manuscript. See also comments to the editor above.

3. There is in fact no demonstration that Thr143 and Thr149 are actually phosphorylated in vivo or in vitro, only that their mutation to Ala diminishes reactivity of cyclin T1 with anti-pThr antibodies. This could be an indirect effect on phosphorylation of a different residue due to conformational changes caused by the mutations. The authors state, on p. 20 of Discussion, that these sites "were missed" in previous studies but to justify that assertion would need to show unambiguously that they are indeed phosphorylated (i.e., not merely the computational prediction in Figure S1 and the circumstantial evidence obtained from mutagenesis studies). Importantly, recombinant, highly active and apparently stable P-TEFb can be purified-and crystallized-in the absence of these phosphorylations (e.g. in Baumli et al., 2008). There is precedent for phosphorylations (on the CDK) being required to stabilize CDK-cyclin complexes in vivo but dispensable in vitro or in overexpression conditions, but to conclude that is the case here would require proof that these residues are indeed phosphorylated in vivo.

We agree and we performed additional studies to confirm the phosphorylation of Thr143 and Thr149 in vivo and in vitro. See answers to comment 2 and to the editor, above. Moreover, examining the literature and talking to direct participants in these crystallographic studies, no-one has been able to express P-TEFb from bacteria. Johnston and Tahirov (as well as Alber and Hurley) groups used P-TEFb purified from baculovirus, where these residues in CycT1 are most likely already phosphorylated.

4. There is a circularity to the logic behind the choice of PKC isoforms to target specifically (p. 13): Thr143 and Thr149 looked like PKC sites, leading the authors to test a panel of PKC inhibitors that were specific for the isoforms they selected for further studies. A more cogent argument would be based on both positive and negative data, i.e., inhibitors of different isoforms (and different kinases entirely) not having the same effects.

Better descriptions are provided in the revised manuscript. Moreover, we did include more PKC inhibitors for comparison, and one non-PKC inhibitor (MEK 1/2 inhibitor) as the negative control. As the new Figures 4B and 4C demonstrate, the MEK1/2 inhibitor had no effect on interactions between CycT1 and CDK9 and Threonine phosphorylation levels of CycT1. In contrast, other more specific PKC inhibitors bisindolylmaleimide IX and HBDDE decreased these interactions and phosphorylation levels. Moreover, the MEK1/2 and other PKC inhibitors were also tested in primary CD4^+^ T cells for further confirmation of our conclusions. As presented in the new figure (Figure 4—figure supplement 1E), the MEK 1/2 inhibitor had no effect on endogenous CycT1 protein levels. Three new PKC inhibitors (not inhibiting PKCα and PKCβ) (enzastaurin, VTX-27 and bisindolylmaleimide IV) also did not decrease CycT1 protein levels. In contrast, other PKC inhibitors (PKCα and PKCβ related) demonstrated consistent effects as presented in Figure 4—figure supplement 1D. Taking all these data together, we can draw the conclusion that phosphorylation of Thr143 and Thr149 in CycT1 is PKC dependent.

5. Selectivity in phosphatase inhibition is notoriously hard to achieve. Nonetheless, there are small molecules with some selectivity for PP1 over PP2A (e.g. tautomycetin), greater selectivity for PP2A (e.g. cytostatin) or ~equal potency towards both (e.g. calyculin A); the authors should try these inhibitors and/or targeted depletion of PP1 catalytic (or regulatory) subunits to test PP1 involvement more rigorously.

In the revised manuscript, we added effects of calyculin A, as suggested. Furthermore, low concentrations of okadaic acid did rule out the involvement of PP2A, compared to the high concentration of okadaic acid and calyculin A (Figures 2A and 2B).

6. The authors should test potentially phosphomimetic substitutions of these residues (e.g. T143E, T149E)-a manipulation they allude to in the Discussion but did not attempt. Although Glu is not guaranteed to mimic pThr, the upside if it does would be considerable: if a constitutive negative charge is sufficient to stabilize P-TEFb, it should increase cyclin T1 in quiescent cells (perhaps conferring sensitivity to Tat) and, if the proposed model is correct, make cyclin T1 in proliferating cells resistant to degradation induced by PKC inhibition.

We indeed tested the phosphomimetic substitutions of Thr143 and Thr149 in CycT1. While T149E restored the binding, neither T143E nor T143D rescued interactions between CycT1 and CDK9. As the reviewer acknowledges, phosphomimetic mutants, particularly those of CDKs, do not always mimic the actual phosphorylation. Molecular dynamics simulations and MM-GBSA binding energy calculations indicates that phosphorylation of Thr143 and Thr149 do have the largest effects in binding stability. Therefore, adding one negative charge on Thr143 might not be sufficient to restore the intra-molecular interaction with Q73. Based on this assumption, we constructed various reciprocal mutant CycT1 proteins and are testing them for their ability to restore P-TEFb assembly without phosphorylation. However, these new substitutions are beyond the scope of the current study and could create a non-degradable P-TEFb complex, which will involve significant further experimentation. We also addressed this point in the response to the editor, above.

7. The study would be greatly strengthened by measurements of CDK9 activity after the various manipulations tested, i.e., cyclin T1 mutations, PKC ablation, PP1 inhibition.

The presented study identifies the mechanism of P-TEFb assembly (CycT1:CDK9 interaction). Since CycT1 unbound CDK9 protein is not enzymatically active, measuring CDK9 activity under these conditions will not be informative.

8. The effectiveness of PKC inhibitors or dominant-negative alleles should be gauged with measurements of phosphorylation on known PKC substrates, if phosphospecific antibodies are available for these targets.

Since PKC inhibitors and dominant negative PKC proteins have been studied extensively, we believe that it is not necessary to repeat them here.

9. The second sentence of the abstract is ambiguously worded: it should be made clearer that what is absent in quiescent cells is P-TEFb, not 7SK RNP (if that is what the authors mean).

Thanks for the corrections. We specified that CycT1 protein levels are greatly diminished in quiescent cells, which leads to the loss of P-TEFb, which does not translate to other components, such as 7SK RNA or HEXIM1.

10. Pp. 2-3, non-expert readers might not appreciate distinction between promoter clearance and promoter-proximal pause release (and thus between functions of CDK7 and -9), which are conflated to a significant degree in the authors' description.

We appreciate the comment. In the revised manuscript, we use the term "promoter-proximal pause-release" throughout, which describes the step of pausing after promoter clearance. Promoter clearance is when RNAPII moves past the promoter after CDK7 had phosphorylated Ser 5, then RNAPII pauses by NELF/DSIF, then promoter-proximal pause-release occurs when CDK9 phosphorylates NELF/DIS and Ser 2.

11. P. 3, first paragraph, sentence starting "CDK9 is…" "DRB" should be "DRB sensitivity inducing factor (DSIF)" (referred to in the next sentence).

We corrected the sentence accordingly.

12. P. 6 and elsewhere, the authors should avoid the nomenclature "T3A" and T2A" to refer to different combinatorial mutations (i.e., of two or three different Thr residues) because it will be confused with single point mutations, i.e., of "Thr3" or "Thr2" (which are nonexistent). I suggest the more conventional shorthand "3TA" and "2TA" to refer to these alleles.

Thanks for the suggestions. We only used the T3A for the triple threonines mutations, and other single or double threonine mutations are named as TT143149AA, T143A and T149A to avoid confusions. We explained this briefly in the first part of results, which is also in the description of Figure 1A.

13. I am just a bit skeptical about the quantification of immunoblot signals: some of the numbers reported for fold-differences do not match the visual evidence (i.e., the band intensities). They appear to be using ECL (rather than fluorescence) for detection and quantification, so these numbers should probably be taken with a grain of salt. I would recommend the authors provide more detail about the quantification, e.g. the raw data for the blot scanning, preferably with error bars.

As we explained to Reviewer #1 (his Figure 2 comment), all WB images were captured by the LI-COR imaging system, and the luminescence was quantified and calculated accordingly. There was no scanning/photoshopping involved in this process. A detailed description of quantifying band intensities of whole lysates, co-IP and IP WB is added in the 'Materials and methods’ of the revised manuscript.

14. There is a sentence fragment on p. 13, starting with "Since the catalytic domains…"

Thank you for pointing out this omission. This sentence fragment has been corrected in the revised manuscript.

15. The precedent of cyclin E degradation cited on p. 20 is interesting but not quite apt: in that case degradation is promoted by phosphorylation of cyclin E (on a sequence motif known as a phosphodegron), and I am not aware of any requirement for cyclin E phosphorylation in binding to CDK2, so its relevance to the proposed mechanism of cyclin T1 degradation is not clear.

In the quoted manuscript, the phosphorylated Cyclin E protein is targeted by SCF E3 ligase complex for degradation. In this section, we wanted to compare the instability of unbound Cyclin E and unstable CycT1 disassociated from CDK9 by its dephosphorylation. This degradation is also important for various cellular functions, as Cyclin E controls the key steps in the cell cycle.

Reviewer #3:The manuscript by Huang and colleagues describes two phosphorylations sites in human Cyclin T1, which regulate the interaction to CDK9 and contribute to the binding and stability of Cyclin T1 in quiescent and terminally differentiated cells. The author describe that the pool of free Cyclin T1 is rapidly degraded when it is not phosphorylated on these two respective sites in the cyclin boxes of cyclin T1. The PP1 phosphates inhibitor okadaic acid leads to P-TEFb stabilization, as less CycT1 is degraded, whereas kinase inhibitors targeting PKC result in P-TEFb disassembly and degradation of CycT1. Using a set of different kinase inhibitors, the authors find that PKC-α and PKC-β are responsible for CycT1 phosphorylation.The findings described in this study bring the transcription regulating cyclins, as CycT1, closer to the cell cycle regulating cyclins, which are long known for the tight regulation between expression upregulation and ubiquitination downregulation. These findings are potentially important, as they could shift our view on the transcriptional cyclins and the need for transcriptional CDK regulators (as 7SK/Hexim1 and Brd4, HIV Tat, AFF4 …), if gene expression and degradation is indeed another layer of regulation of these cyclins.There is one major conundrum to this study: Multiple biochemical and structural studies on CDK9 and Cyclin T1 have been performed, often by co-expression and co-purification of these two subunits. In none of them (to my knowledge) a phosphorylated T143 or T149 residue was identified. The sentence "Unphosphorylated CycT1 dissociates from P-TEFb and is degraded." (page 20 and similar in the abstract) seems therefore farfetched. If pT143 and pT149 stabilizes the complex in such a profound degree, why are researchers able to form the non-phosphorylated complex in the first place? CycT1 can be expressed from *E. coli* exhibiting no phosphorylation and binds properly well to CDK9. It would be very beneficial and in support of this study to know the dissociation constants for the interaction partners in a CycT1 phosphorylated and non-phosphorylated state.

As described in the answer to Reviewers 1 and 2, all P-TEFb structures (including the studies by Baumli 2008, Tahirov 2010, Alber 2013, Alber 2014) were obtained after the purification of the complex from the eukaryotic (insect S9 cells) cells, where CDK9 and CycT1 are likely phosphorylated. Also, enzymatically active P-TEFb complex can be purified from eukaryotic cells, but not from *E. coli* where post-translational modifications (PTM) are not preserved. Although structural studies by Geyer group used CycT1 expressed in *E. coli*, these studies did not include CDK9. In fact, there are no studies with P-TEFb expressed from *E. coli*. This finding has been confirmed by our direct communications with Prof. Price, Alber and Hurley.

In addition, the new table S1 demonstrates that after further Molecular dynamics simulations and MM-GBSA binding energy calculations, the phosphorylation of Thr143 and Thr149 in CycT1 results in a thermodynamically more stable CycT1:CDK9 association. The predominant contribution to the increased binding energy of the complex comes from electrostatic (ΔE_elec_) and polar solvation (ΔE_solv−polar_) energies, which is consistent with the previous co-IPs between CycT1/mutant CycT1 and CDK9/mutant CDK9 proteins (Figures 3B, 3C and 3D). Furthermore, new Figure 2D demonstrates that with okadaic acid treatment, total phosphorylation levels in purified WT CycT1(280) are much higher than in its purified mutant CycT1(280)TT143149AA protein after phospho-staining. Taken together, Thr143 and Thr149 in CycT1 are actually phosphorylated in vivo, and such phosphorylation is necessary for P-TEFb assembly in cells. See also additional comments to the editor and Reviewers 1 and 2.

I am surprised that a phosphorylated threonine mimetic, as glutamic acids (E) or aspartic acid (D), was not analyzed in these experiments. To this reviewers' opinion the authors should analyze how a double mutation of CycT1 (T143E/T149E) acts on CDK9 binding, P-TEFb complex formation and the stability of Cyclin T1. Using only the T143A/T149A mutant but not a phosphorylation mimetic is a lack in the design of the study.

Please see our answer to the editor and Reviewers 1 and 2.

Throughout the entire manuscript, the authors are unprecise and ambiguous with predictions (computational, structural, …) and 'real' experimental findings. E.g.: Page 18, Discussion, 3 sentence: "Structural analyses revealed that phosphates on Thr143 and Thr149 in CycT1 increase intramolecular and intermolecular binding to specific residues in CycT1 and CDK9, respectively, which potentiates P-TEFb assembly and stabilizes CycT1." In Figure 3A, a model is displayed but not an experimentally determined structure with phosphorylated residues, as 3MI9 does not contain any phosphorylation in Cyclin T1. The figure legend to 3A should clearly say this. The description in the main text is okay, but the text of the figure legend is clearly misleading and almost fraud. This is a model only!Another example is on page 8: "An online database for phosphorylation site prediction (NetPhos 3.1, developed by Technical University of Denmark) scores Thr143 and Thr149 above the threshold value (default 0.5), indicating that these threonines are potential phosphorylation sites (Figure S1). To verify that Thr143 and Thr149 are the main phosphorylation sites in CycT1, …" The second sentence starting with "To verify …." suggests, that the first observation are real data but not simply predictions. Moreover, the prediction has been made on a 43-residue sequence but not the full CycT1 structure (Figure S1) as I understand. This does not account for any tertiary structure etc. and is therefore not to the highest standard. I would not consider this analysis good evidence for phosphorylation sites.

We apologize the confusion. In the revised manuscript, we clarified the prediction and actual experimental findings. First of all, the description of Figure 3A is changed greatly and we emphasized that this is merely a model to provide a better understanding of which residues in the CycT1:CDK9 complex are targeted by Thr143 and Thr149. We also added more predictions based on molecular dynamics simulations and MM-GBSA binding energy calculations (Figure 3A and Table 1), which further indicate that the phosphorylation of Thr143 and Thr149 in CycT1 results in a thermodynamically more stable CDK9:CycT1 association. The predominant contribution to the increased binding energy of the complex comes from electrostatic (ΔE_elec_) and polar solvation (ΔE_solv−polar_) energies. Results of following IP-WBs support the model (Figures 3B, 3C and 3D). In addition, the NetPhos search was first a prediction, not providing actual evidence, but subsequent experiments using mutations in these sites verified this prediction. In the revised manuscript, we made this point clearer.

There are clearly dissociation constants missing for the statement that the threonine phosphorylation in cyclin T1 is increasing its binding capacity to CDK9 and the stability of the cyclin. From the model it seems that pT143 could also only act indirectly as it forms internal contacts within the cyclin but is not in the direct interface to the kinase.

According to the molecular dynamics simulations and MM-GBSA binding energy calculations, the phosphorylation of Thr143 stabilizes intramolecular interactions with Gln73 in CycT1. Our interpretation is that the stabilization of intramolecular interaction is necessary to present an appropriate conformation of CycT1 to form the high-affinity binding surface to CDK9 where phosphorylated Thr149 can be located closely to its target Lys69 in CDK9. The co-IPs in Figures 1 and 3 indicate that these interactions between CDK9 and mutant CycT1 T143A or CycT1 T149A proteins are weaker than those with the WT CycT1, but the double mutant CycT1 TT143149AA protein reveals a synergistic, rather than additive, impact on CDK9 binding. Taken together, although it is not located in the direct interface with CDK9.

[Editors' note: further revisions were suggested prior to acceptance, as described below.]

Reviewer #2:This is a revision of a manuscript I reviewed previously. The authors have addressed one of my major concerns by providing evidence that PKC can phosphorylate a cyclin T1 fragment in T143/149-dependent fashion in vitro (Figure 5G). I am less persuaded by the evidence implicating PP1 in dephosphorylating cyclin T1, but this is largely owing to the technical difficulty of proving phosphatase specificity in cell-based experiments. Taken together, there is substantial reason to believe, based on the data presented, that presence or absence of PKC-dependent signaling is a determining factor in P-TEFb activity levels in different settings (e.g. resting or exhausted versus activated T cells).

We added additional phosphatase inhibitors in the revised manuscript. Examining the literature, we find that experimental procedures followed in our work represent the accepted gold standard. The plethora of PP1 and PP2 subunits renders most other approaches difficult if not impossible.

My biggest concern remains, however, that the mechanism the authors propose for this regulation-phosphorylation of two specific residues on cyclin T1 that stabilize an interaction with CDK9-is only incrementally closer to being validated in the revised version. The new data addressing this point were obtained in a complicated setup, in which a severely truncated cyclin T1 fragment (missing residues needed for interaction with CDK9) was expressed in 293T cells and then pulled down and detected in gels by a commercial phosphoprotein-staining reagent, either intact (Figure 2D) or after tryptic digestion (2E). Mutation of T143 and T149 to A abolished the signal, which is promising but still preliminary evidence that these sites might be phosphorylated in intact cyclin T1 under physiologic conditions. This does not address the concern, also expressed by Reviewer 3, that this requirement-and the phosphorylations themselves-have escaped previous detection. The authors dismiss this concern by pointing out that structures of P-TEFb were solved for complexes expressed in insect cells, but this is not persuasive; in those structures, phosphorylation on T186 of the CDK9 activation segment was readily observed (Baumli et al., 2008; Tahirov et al., 2010). Fully active, stable P-TEFb complexes can also be reconstituted from purified CDK9 expressed in insect cells and cyclin T1 purified from bacteria. Finally, a quick check of the PhosphoSite online data base did not turn up either of these sites.

We agree that the phosphorylation of T186 was observed in the crystal structure of P-TEFb, but important others in CDK9 and CycT1 were not. In addition, CycT1 from position 1-280 is fully functional in cells when presented to RNA polymerase via RNA (many papers from our and other groups). The full length CycT1 1-726 is only required for DNA presentation. There, the histidine-rich stretch in CycT1 binds to the CTD and directs the modification of RNA polymerase from the distance. In our 2002 manuscript, we discovered that promoters for the most part initiate transcription and enhancers elongate transcription, precisely because of this extended C-terminal half of CycT1. Thus, CycT1 1-280 is a fully functional P-TEFb when presented to RNA polymerase as it begins to synthesize RNA. That is precisely how Tat works! However, CycT1 1-280 has two almost identical large tryptic digests, which necessitated a further truncation to create a unique, identifiable fragment that is phosphorylated by the gold standard for the direct mapping of phorphorylated residues in proteins, that of comparing wild type and mutated peptides for phosphorylated residues. As explained in greater detail below, it would have been impossible to demonstrate the phosphorylation of these adjacent threonines in any other way than with a truncated, validated CycT1 protein. We went to great lengths to explain this logic in the revised manuscript.

To the second point, no one has been able to reconstitute P-TEFb from bacterially expressed components. We contacted Alber/Hurley and Tahirov groups, especially those responsible for generating P-TEFb in insect cells and they confirmed this finding. Importantly, they also investigated only P-TEFb lacking the C-terminal half of CycT1, which is unstructured and resisted all attempts at crystallization. Other validated (published) phosphorylated residues in CDK9 also did not appear in these structures.

There is a suggestion throughout this review that we did not follow precedent. The only reason this study was undertaken is because of two publications, which revealed that T143, T149 and two other residues in CycT1 were essential for P-TEFb. In our manuscript, we could eliminate contributions of these 2 other residues, which left T143 and T149. Several independent prediction protocols, all freely available on line (to all investigators) revealed that they are bona fide phosphorylation sites, especially for PKC.

In separate work by us and others, we found that CycT1 was degraded in resting and terminally differentiated cells, which was prevented by MG132 and bortezomib (bortezomib or Velcade^R^ is a proteosomal inhibitor that revolutionized the treatment of multiple myeloma). In these studies, they played key roles in reversing HIV latency, especially when combined with PKC agonists. Thus, our findings are not new to us but follow abundant precedent in the literature!

So I must reiterate my initial position that, because the authors are proposing a novel and highly detailed mechanism for regulating an essential transcription regulator, the existence, necessity and sufficiency of T143 and T149 phosphorylation simply must be proven and not merely inferred. Typically, a study such as this would include dispositive supporting data obtained via mass spectrometry, phospho-specific antibodies, or both, to gain acceptance at a top-tier journal.

Please note that mass spectrometry was never mentioned in the initial review. We attempted to make specific antibodies against phosphorylated T143 and T149. However, extensive discussions with experts in making anti-phosphorylated peptide antibodies at UCSF and commercial sources, led to the conclusion that 1. these phosphorylated residues reside in a deep and cavernous pocket (where CDK9 binds) and 2. are spaced close together, which would be difficult if not impossible to access and distinguish by rather bulky antibodies. As to mass spectrometry, we consulted two experts as well. Upon reading the paper, they thought our data were convincing, internally consistent and exhaustive. They pointed out that mass spectrometry is notoriously bad with two adjacent (closely positioned) phosphorylated residues. Agreeing that sampling of free and complexed CycT1 (unphosphorylated and phosphorylated) would present another challenge of abundance and representation. Another problem was with sample preparation and removal of phosphate groups during isolation and purification of P-TEFb. Intrinsic to mass spectrometry, they mentioned acidity, competition for ionization, suppression of signals, unidentifiable peaks and incomplete peptide maps. There are many other considerations, too many to enumerate here (but we are happy to provide additional statements and references). Some relate directly to the published data on CycT1, where only 2 phosphorylated sites were observed in the full-length protein (none in crystal structure of P-TEFb). Using traditional mapping of phosphorylated residues by radiography, Jones and Zhou groups found many more sites in CDK9 and CycT1. Our Figure 2 reveals that there are also additional sites in the truncated CycT1. In conclusion, both experts urged strongly that any mass spectrometric findings would have to be validated by other methods, which they thought we did already. Indeed, they were puzzled that these other accepted and validated measures were not accepted by this reviewer.